

# Observing atmospheric rivers using multi-GNSS airborne radio occultation: system description and data evaluation

Bing Cao[1], Jennifer S. Haase[1], Michael J. Murphy, Jr.[1,2], and Anna M. Wilson[3]

[1]Institute of Geophysics and Planetary Physics, Scripps Institution Oceanography, University of California San Diego, La Jolla, California, USA
[2]Now at: Global Modeling and Assimilation Office, NASA Goddard Space Flight Center, Greenbelt, Maryland, USA, and GESTAR-II, University of Maryland Baltimore County, Baltimore, Maryland, USA
[3]Center for Western Weather and Water Extremes, Scripps Institution Oceanography, University of California San Diego, La Jolla, California, USA

**Correspondence:** Jennifer S. Haase (jhaase@ucsd.edu)

**Abstract.**

Atmospheric Rivers (ARs) are narrow filaments of high moisture flux responsible for most of the horizontal transport of water vapor from the tropics to mid-latitudes. Improving forecasts of ARs through numerical weather prediction (NWP) is important for increasing the resilience of the western US to flooding and droughts. These NWP forecasts rely on the improved

understanding of AR physics and dynamics from satellite, radar, aircraft, and in situ observations, and now airborne radio occultation (ARO) can contribute to those goals. The ARO technique is based on precise measurements of Global Navigation Satellite Systems (GNSS) signal delays collected from a receiver onboard an aircraft from setting or rising GNSS satellites. ARO inherits the advantages of high vertical resolution and all-weather capability of spaceborne RO observations and has the additional advantage of continuous and dense sampling of the targeted storm area. This work presents a comprehensive ARO

dataset recovered from four years of AR Reconnaissance (AR Recon) missions over the eastern Pacific. The final dataset is comprised of $\sim 1700$ ARO profiles from 39 flights ($\sim 260$ flight hours) from multiple GNSS constellations. Profiles extend from aircraft cruising altitude (13–14 km) down into the lower troposphere, with more than 50% of the profiles extending below 4 km, below which the receiver loses or cannot initiate lock. The horizontal drift of the tangent points that comprise a given ARO profile greatly extends the area sampled from just underneath the aircraft to both sides of the flight track (up to $\sim 400$ km).

The estimated refractivity accuracy with respect to dropsondes is $\sim 1.2\%$, in the upper troposphere where the sample points are closely collocated. For the lower troposphere, the agreement is within $\sim 7\%$ which is the level of consistency expected given the nature of atmospheric variations over the 300–700 km separation between the lowest point and the dropsonde.

## 1 Introduction

Atmospheric Rivers (ARs) are narrow plumes of concentrated moisture that transport large amounts of water vapor over long

distances. These moisture plumes can stretch for thousands of kilometers and can be as wide as a few hundred kilometers. They are often associated with extratropical cyclones (ETC) and develop over the ocean, impacting the west coasts of continents at



mid-latitudes (Zhu and Newell, 1994; Ralph et al., 2004, 2018; Zhang et al., 2019). ARs play an essential role in the global water cycle, by transporting water vapor between the tropics and mid-latitudes. ARs contribute more than 90 % of the meridional water transport within 10 % of the earth circumference at mid-latitudes (Zhu and Newell, 1998). On the United States (US) west coast, they contribute about 30–50 % of the annual precipitation and are the major cause of extreme precipitation events (Dettinger, 2013). On the one hand, they can bring much-needed precipitation that can contribute to the water supply and alleviate droughts; on the other hand, the heavy rainfall ARs bring over an extended period of time can cause severe flooding, resulting in fatalities and large economic losses (Corringham et al., 2019; Ralph et al., 2020). Therefore, a significant effort in terms of field campaigns, numerical simulations, and data assimilation experiments has been dedicated to the investigation of ARs, in order to better understand the physics and dynamics that characterize them as well as improving forecasts.

Challenges exist in forecasting the landfall location and intensity of ARs (Lavers et al., 2020; Stewart et al., 2022; Cordeira and Ralph, 2021), as they are poorly observed when they originate and evolve over the mid-latitude oceans where direct observations are sparse. Satellite radiance, atmospheric motion vectors (AMVs), and integrated water vapor (IWV) have helped improve forecasts of ARs to some extent, with their dense horizontal sampling, however their vertical resolution is poor. Moreover, satellite observations in ARs typically have limited coverage and increased errors due to their sensitivity to clouds and heavy precipitation, which are quite common in the AR environment. Zheng et al. (2021) quantitatively described the existence of a data gap in the northeast Pacific in the cloudy regions of ARs that satellite observations are unable to adequately fill from near the surface to the middle troposphere, hence the motivation for reconnaissance observations. Experiments using adjoint models identified the sensitivity of uncertainties in the precipitation from landfalling ARs to initial condition errors in the humidity and other atmospheric variables inside ARs (Doyle et al., 2014; Reynolds et al., 2019), which underlines the importance of direct observations of humidity in the AR environment.

Compared to satellite observations, measurements taken from reconnaissance aircraft that densely sample the target areas at desired times have an advantage for synoptic-scale to mesoscale systems such as ARs and tropical cyclones (TCs). These extreme events are associated with highly variable environments within a relatively small area. The AR Reconnaissance (AR Recon) program is a collaborative effort of the Center for Western Weather and Water Extremes (CW3E) and the National Oceanic and Atmospheric Administration (NOAA) National Center for Environmental Prediction (NCEP), involving several domestic and international partners, that focuses on atmospheric dynamics, the predictability of ARs, airborne instrumentation, and data assimilation in numerical weather modeling. The primary airborne observations come from dropsondes, which directly measure pressure, temperature, moisture, and winds beneath the flight track. AR Recon grew out of the California Land-falling Jets Experiment (CALJET) and CalWater field campaigns in the early 2000s (Ralph et al., 2005, 2016) when aircraft flights were combined with coordinated observations, such as upslope moisture flux from soundings and wind profilers acquired on land, to quantify the relationship between ARs and orographic precipitation. AR Recon is an expanded effort that includes three aircraft flying each year: one Gulfstream-IV provided by NOAA, and two WC-130J aircraft provided by the US Air Force (USAF). The first AR Recon campaign was carried out in 2016 and has become part of the National Winter Season Operations Plan (NWSOP), occurring every year since 2018. The dropsondes are assimilated into operational NWP systems, and the information collected is also being used in research studies to further understand the dynamics and processes that are





the main drivers of key AR characteristics, such as strength, position, length, orientation, and duration. Positive impacts of dropsondes on NWP forecasts of precipitation from ARs have been found (Stone et al., 2020; Lord et al., 2023), particularly from multi-day sequences of flights (Zheng et al., 2021), and in combination with drifting buoys (Centurioni et al., 2017; Reynolds et al., 2023). The dropsondes also improve the impact of satellite radiance data in AR forecasting through bias correction (Zheng et al., 2022).

Airborne GNSS radio occultation (ARO) is a remote sensing technique complementary to dropsondes on AR Recon flights. The retrieved ARO refractivity profiles combined with dropsonde observations collected over the otherwise data-sparse ocean are intended to be part of the solution to improve AR forecasting. ARO was first implemented as a proof-of-concept in the Pre-Depression Investigation of Cloud-Systems in the Tropics (PREDICT) field campaign in 2010, during which the GPS receivers were deployed on the National Science Foundation (NSF) G-V aircraft (Haase et al., 2014; Murphy et al., 2015). The ARO dataset was assimilated in a study of 2010 hurricane Karl (Chen et al., 2018). Results show a clear impact of ARO observations on the forecast sea level pressure errors. ARO observations have been an essential component of AR Recon since 2018, and the technique has undergone many advances over the years. Haase et al. (2021) described ARO observations from the first flights in the 2018 AR Recon campaign, and highlighted some of the advances, for example, the first RO profile that was retrieved from the European Galileo constellation. A preliminary data assimilation (DA) experiment revealed a noticeable increment in moisture in areas not sampled by dropsondes. Analysis of the suite of AR Recon observations and their impact on forecasting is ongoing, including developing specialized ARO assimilation methods (Hordyniec et al., 2024).

The ARO observation technique follows the same principles as spaceborne GNSS radio occultation (abbreviated here as SRO to distinguish it from ARO). The ability of SRO to describe high moisture features over the oceans was demonstrated in a comparison of the Special Sensor Microwave Imager (SSM/I) integrated water vapor (IWV) products with independent IWV estimates from COSMIC RO products in the eastern Pacific (Wick et al., 2008). They showed strong agreement with nearly zero mean bias and a 3 mm RMS difference in precipitable water. Ho et al. (2018) also found these two types of satellite products are highly consistent on a global scale over the long term, with less than 2 mm difference in clear sky conditions with no precipitation. This capability demonstrated the potential for SRO to resolve the fundamental moisture features of ARs. Neiman et al. (2008) used composite SRO profiles from the COSMIC constellation over two days to describe the general features of the greater AR environment, in terms of vertical distribution of moisture and temperature. The SRO profiles in the lower troposphere showed meteorologically consistent vertical structures in temperature and moisture within an AR compared with other observations and reanalysis models. Due to the sparsity of the observations this was only possible in a composite sense accumulated over many days. SRO and dropsondes observations collected over three years of AR Recon were examined by Murphy and Haase (2022), who found deeper penetration of SRO profiles within ARs due to the relatively small gradients of refractivity in the vertical compared to the environment surrounding ARs. This highlights the potential for RO to sample the hard-to-reach areas that results with adjoint models suggest have the most impact on the forecasts of ARs (Doyle et al., 2014; Reynolds et al., 2019).

SRO observations are now routinely assimilated into operational NWP systems at organizations such as the European Centre for Medium-Range Weather Forecasts (ECMWF) and the National Centers for Environmental Prediction (NCEP). The positive





impact of the data in the upper troposphere and lower stratosphere has been studied and confirmed in global models (Ruston et al., 2022, and references therein). It has been more of a challenge to demonstrate impact in the lower troposphere when assimilating RO observations in the highly variable mesoscale environments of extreme events such as in TCs and ARs. This is

partly due to the limited and quasi-random sampling available over the active area of interest. Many studies have investigated and found positive impacts from assimilating SRO data for improved tropical cyclone track and intensity prediction (Ma et al., 2009; Liu et al., 2012; Chen et al., 2015). Usually the impact was attributed to a few key profiles being available in the vicinity of the storm. Ma et al. (2011) assimilated SRO profiles from COSMIC and CHAMP for a 24-hour forecast for a key AR event. A total of 433 SRO profiles from 7 days were assimilated, which improved the moisture analysis and resulting forecast. In these

earlier studies, SRO profiles were generally so coarsely distributed that data had to be accumulated over a longer period and/or larger area, given that these profiles were rarely in the high-sensitivity region of a TC or AR. The recent launch of COSMIC-2 with six low-inclination (24°) satellites yields more profiles, particularly in the tropics, which substantially increases the sampling for TCs. Miller et al. (2023) assimilated the COSMIC-2 SRO bending angle for six 2020 Atlantic hurricane cases. An average of 3 profiles within the highest resolution domain (11° by 11°) per cycle (6 hours) are assimilated, which yields a

modest 10% intensity forecast skill improvement for several lead times. Numbers are increasing in the mid-latitudes with the launch of recent commercial satellite constellations, however ARO focuses on the localized storm environment, with a dense distribution of observations, so it is more likely to capture a sensitive area that could impact the downstream evolution of a particular storm event.

      The past decade has seen significant advances in GNSS technology, including the launch of new GNSS satellites, the com-

pletion of new GNSS constellations such as Beidou, the development of new signals at additional frequencies, and advances in receiver technology and tracking algorithms. These factors expand the general capability of GNSS and provide a specific advantage to RO because stronger GNSS signals and more occultations are recorded. Haase et al. (2014) showcased the feasibility of performing RO observations from an aircraft platform, and Haase et al. (2021) demonstrated the potential utility of ARO for observing atmospheric rivers in the AR Recon 2018 campaign. That study evaluated the data quality by comparing

individual profiles with dropsondes and reanalysis products and demonstrated the capability of the ARO data to resolve AR features using a simple DA experiment. Since then, the quantity of ARO data has greatly increased and more data has been accumulated, such that it is possible to do a comprehensive statistical study that surpasses earlier work. The scope of this study is to describe the hardware and software of the ARO technology, including the receivers, antennas, and retrieval algorithms used to process raw data to retrieve thermodynamic profiles. It also provides a comprehensive description and assessment of

the 4-year ARO dataset from AR Recon. ARO observations have some unique characteristics; some are strengths, and some are limitations. They are analyzed and discussed in depth in this study, aiming to provide useful guidance to any users of the ARO datasets and facilitate the further deployment of ARO systems on additional aircraft. With that objective in mind, the study is organized with the following structure. Section 2 describes the basics of ARO observation systems and summarizes the retrieval procedures. Section 3 describes the important sampling characteristics of the ARO profiles. Section 4 presents the

data quality through comparisons with independent data and evaluates the accuracy of the products. Section 5 and section 6 provide a summary and perspectives on the future exploitation of the system.





## 2 Methodology and instrumentation

### 2.1 Airborne radio occultation

When radio waves propagate through a layered atmosphere, they will deviate from a straight line by a bending angle and be

delayed by a small amount of time depending on the refractive index of the atmosphere (Fig. 1). The atmospheric refractivity, $N$, for radio waves at GNSS frequencies in the neutral atmosphere is described by

$$N = (n-1) \times 10^6 = 77.6\frac{p}{T} - 6.3938\frac{p_w}{T} + 3.75463 \times 10^5 \frac{p_w}{T^2}, \tag{1}$$

where $n$ is the refractive index, $p$ is atmospheric pressure [in hPa], $p_w$ is water vapor pressure [in hPa], and $T$ is atmospheric temperature [in K], respectively (Rüeger, 2002).

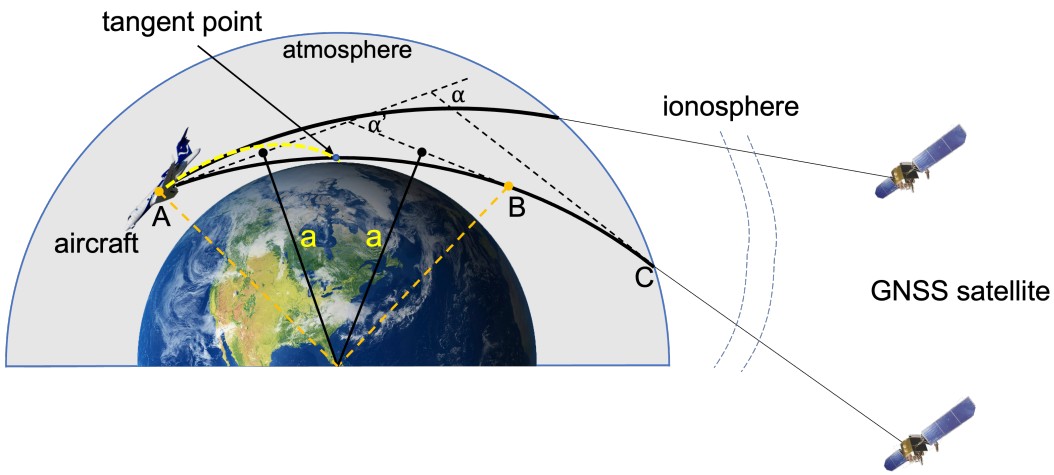

**Figure 1.** Schematic diagram illustrating airborne radio occultation. The solid lines are ray paths of navigation signals, transmitted by GNSS satellites and continuously tracked by the receiver onboard an aircraft. The GNSS antenna receiving the navigation signal is installed on the top of the fuselage, and receives the signal sideways from any unobstructed direction. The full bending angle, $\alpha$, is the refractive bending accumulated along the ray path over segment $AC$ as the signal propagates through each successive atmospheric layer. Point $B$ on the ray path is symmetric to the aircraft position $A$ with respect to the tangent point. The bending angle $\alpha'$ is the partial bending angle that accounts only for the bending of the ray path along segment $AB$, and $a$ is the corresponding impact parameter. The segment of the ray path above $C$ passes through the dispersive ionosphere. The retrieved slant profile is comprised of refractivity values at a series of tangent points, one for each ray path as the GNSS satellite sets, and is indicated by the curved yellow dashed line.

As shown in Fig. 1, the curved segment $AC$ along the ray path is within the neutral atmosphere, typically below 80 km

altitude. The segment of the ray path above $C$ undergoes the influence of the ionosphere, however, the ionospheric effects are removed using the dual-frequency combination of observed excess phase, so the corresponding bending is not considered. The point on the ray path that is closest to the Earth's surface is called the tangent point. Following the relative motion of aircraft





and satellites, the ray path scans the atmosphere from top to bottom or vice versa. At one moment in time, the aircraft, satellite, and center of the Earth define a plane containing the tangent point, called the occultation plane. Because the velocity vectors
of aircraft and satellites are not in exactly opposite directions, the orientation of the occultation plane varies slightly over one occultation.

The refractive bending of segment $AC$ in an assumed one-dimensionally varying layered atmosphere is described by an integral over radius from the center of curvature of the Earth

$$\alpha(a) = 2a \int_{r_t}^{r_R} \frac{1}{n} \frac{dn}{dr} \frac{dr}{\sqrt{n^2 r^2 - a^2}} + a \int_{r_R}^{r_T} \frac{1}{n} \frac{dn}{dr} \frac{dr}{\sqrt{n^2 r^2 - a^2}}, \quad (2)$$

where $r_R$, $r_T$ and $r_t$ are the radius of receiver (aircraft), transmitter (satellite) and tangent point on the ray path, respectively.
$a = n_t r_t$ is the impact parameter for this ray path (Fjeldbo et al., 1971).

The bending angle can be derived from the observed Doppler shift and position and velocities of aircraft and satellites (Vorobev and Krasilnikova, 1994). There is a large difference between ARO and SRO, where the receiving LEO satellites are outside the atmosphere at higher orbits. The aircraft is flying within the atmosphere, leading to the asymmetric geometry of the segment $AC$. In order to utilize the Abel transform to retrieve the refractive index, a correction is needed to determine the
bending of the symmetric part of the ray path, denoted by $AB$. The correction requires knowledge of the refractivity above the receiver to estimate the bending of the segment of the ray path $BC$, which is not possible in practice. An alternate approach is to determine the bending of the ray path with the same impact parameter at a positive elevation angle from point $A$ that is approximately equal to the bending of segment $BC$. The partial bending angle is defined as the difference between the total bending accumulated for the ray path arriving at the receiver from an elevation angle below the horizon (negative elevation
angle) (Eq. 2) minus the bending from the ray path from positive elevation angles at the same impact parameter (second term in Eq. 2) (Xie et al., 2008). It approximately corresponds to the accumulated bending from the symmetric ray path segment below the receiver altitude (first term in Eq. 2 and $AB$ in Fig. 1). The corresponding partial bending angle $\alpha'$ is then inverted using the Abel transform to retrieve the refractive index profile.

$$n(a) = n_R \cdot \exp\left(\frac{1}{\pi} \int_a^{n_R r_R} \frac{\alpha'(x)\, dx}{\sqrt{x^2 - a^2}}\right) \quad (3)$$

where $n_R$ is the refractive index at the aircraft location. $n_R$ can be calculated from aircraft flight-level meteorological measure-
ments using Eq. 1. The refractive index profile as a function of altitude $n(h)$ is retrieved using the relation $a = nr = n(R_c + h)$, where $R_c$ is the local radius of curvature of the ellipsoidal Earth and $h$ is the altitude above the ellipsoid.

The temperature, water vapor pressure, and pressure cannot be determined independently from refractivity using Eq 1. In the upper troposphere above $\sim 7$–9 km where the moisture can be neglected, the temperature and pressure can be estimated independently with the assumption of hydrostatic equilibrium (Kursinski et al., 1997). However, in the lower troposphere where
the contribution of moisture is significant, additional information is required from other sources, such as weather models. Then, the moisture contribution can be separated from the hydrostatic term using variational or other optimization methods.



As mentioned above, the retrieved refractivity, with its combined effects of the hydrostatic and moisture terms, can be used directly to constrain NWP models through variational data assimilation.

## 2.2 ARO deployment on the NOAA G-IV Jet

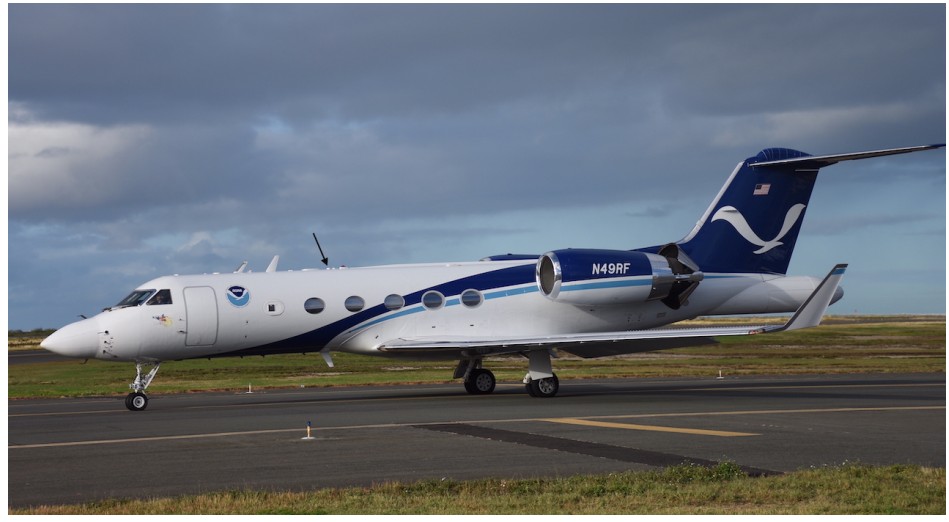

**Figure 2.** NOAA G-IV aircraft (registration # N49RF) seen on the taxiway of Honolulu Daniel K. Inouye International Airport (HNL) before departure for an AR Recon flight on January 31, 2021. The black arrow marks the location of the science GNSS antenna installed on top of the fuselage.

NOAA's Gulfstream IV-SP (G-IV) is a high-altitude, high-speed platform that is one of the aircraft deployed for AR Recon missions. Since 2018, it has been equipped with ARO instrumentation. Up to eight-hour endurance and 6700 km range make it an ideal platform to track fast-evolving storms. The primary equipment on the G-IV is the Airborne Vertical Atmospheric Profiling System (AVAPS). The system releases dropsondes from the aircraft that descend with a parachute, continuously measuring the state of the atmosphere (pressure, temperature, moisture, winds) and creating a nearly vertical profile underneath
the flight track. The G-IV also carries an X-band Tail Doppler Radar (TDR), measuring the precipitation and winds and providing information about the convective activity surrounding the flight track. Multiple meteorological sensors are installed outside the aircraft (nose cone) to measure ambient temperature, dynamic and static pressure, and water vapor mixing ratio at flight level. These in situ measurements are used in the ARO retrieval process. There are two GPS antennas installed on top of the aircraft fuselage; one is dedicated to navigation and one is the science GPS antenna for the AVAPS system, which provides
time synchronization and location required for the dropsonde launch. During a typical AR Recon flight, the G-IV jet was flying across the AR at a speed of 800 km/h at altitudes over $\sim$ 13,000 m during which dropsondes were released along the track and the TDR was sensing the precipitation in the region surrounding the aircraft. The flights lasted about 7–8 hours, during



**Table 1.** Receivers and antennas used in the AR Recon campaigns onboard the NOAA G-IV.

| Year | Receiver Model | Antenna Model | Antenna Type | Tracked Signals |
|------|----------------|---------------|--------------|-----------------|
| 2018 | PolaRx5[a] + ROC2 + POS AV | AT2775-80[b] | GPS only, L1/L2 | GPS + Galileo |
| 2019 | ROC2[a] | AT2775-80 | GPS only, L1/L2 | GPS + Galileo + GLONASS |
| 2020 | AsteRxU[a] | AT2775-80 | GPS only, L1/L2 | GPS + Galileo + GLONASS |
| 2021 | AsteRxU[a] + Pwrpak7 + GSS6450 | AT1675-180[b] | GNSS/L-band, L1/L2/L5 | GPS + Galileo + GLONASS + L-band |

[a] Underlines mark the receivers from which the data were post-processed and presented in this work.

[b] Both antennas are manufactured by AeroAntenna Technology, Inc.

which about 20–30 dropsondes were released usually over the core of the AR at an interval of $\sim$ 8–10 min, corresponding to a separation of $\sim$ 100–150 km.

## 2.3 GNSS receivers and antennas

Over the four years (2018–2021) of AR Recon flights, multiple GNSS receivers were installed and tested onboard the G-IV aircraft. The original GPS-only science antenna on the aircraft was upgraded to full GNSS capability before AR Recon 2021. Table 1 lists the receivers used in the past AR Recon missions and the antennas installed on the G-IV aircraft.

The PolaRx5 (deployed in 2018) and AsteRxU (deployed each year since 2020) are both commercial off-the-shelf geodetic grade GNSS receivers manufactured by Septentrio and widely used for high-accuracy positioning in seismology, geodesy, and meteorology. ROC2 is a low-cost very light-weight GNSS receiver built at Scripps Institute of Oceanography for RO observations from long duration super-pressure balloons in the Strateole-2 campaign (Haase et al., 2018; Cao et al., 2022). It contains a Septentrio AsteRx4 OEM board tracking GNSS signals on multiple frequencies from all major constellations, with the capacity to log data from two antennas. The ROC2 receiver was deployed on the G-IV aircraft in 2018 for testing and to assess instrument precision (Haase et al., 2021). In 2019, a ROC2 was deployed to perform ARO observations on the G-IV as a piggyback on the NOAA Gravity for the Redefinition of the American Vertical Datum (GRAV-D) mission. All of these receivers perform phase-locked loop (PLL) tracking of the GNSS signals. The Applanix POS AV deployed in 2018 is a GNSS/Inertial Navigation System (INS) system and a sub-component of the prototype GNSS Instrument System for Multistatic and Occultation Sensing (GISMOS) instrument (Garrison et al., 2007). It provides Inertial Measurement Unit (IMU) aided high-accuracy kinematic positioning solutions (Muradyan et al., 2011). The Novatel Pwrpak7 is also a GNSS/INS system and is part of the micro-gravimeter system for the GRAV-D missions operated by NOAA/NGS. Several GRAV-D survey flights were conducted by the G-IV in the downtime during AR Recon 2021 in Hawaii. We took advantage of the deployment to test the real-time Precise Point Positioning (PPP) correction service (TerraStar-C) for the Pwrpak7, which receives correction signals broadcast by geosynchronous satellites and uses inputs from a Honeywell IMU to generate positions with improved accuracy in real-time. In 2021, an experimental raw RF signal recorder (Spirent GSS6450) was also installed on the aircraft to record raw digitized GNSS signals on L1 frequency at a rate of 10 MHz. The data were recorded for future post-processing





using an open-loop (OL) tracking algorithm (Muradyan, 2012; Wang et al., 2017) to sample the lower moisture troposphere where the conventional PLL receivers would not be able to track.

The original antenna (AT2775-80) used by the AVAPS system on the G-IV was a GPS-only antenna designed for L1 and
L2 GPS frequencies. However, our tests showed that the frequency coverage actually extends to a broader range, enabling the tracking of Galileo (E1 and E5b) and GLONASS (G1 and G2) signals, but with a relatively low gain. In 2018, we configured the receiver to track and log data from GPS and Galileo satellites as their signal frequencies (L1 and E1, L2 and E5b) are relatively close, after which we confirmed the PolaRx5 and ROC2 receivers both picked up good recordings of Galileo signals from this antenna. Starting in 2019, we implemented GLONASS signal tracking for full GNSS operation. By tracking signals
from all three constellations, more than double the ARO profiles were retrieved within a particular area and time window, than when GPS-only data were logged. A splitter was used to share the RF feed from the antenna to both the AVAPS and ARO systems at the cost of some signal strength loss (roughly 3.5 dB). This GPS-only antenna was upgraded to a new multi-GNSS antenna (AT1675-180) at the end of 2020. The newly upgraded antenna now has full coverage of all GNSS signal frequencies and the capacity for receiving a positioning correction service on L-band if desired. The installation location of this antenna is
marked by the arrow in Fig. 2.

This study presents post-processed results using data from PolaRx5 in 2018, ROC2 in 2019, and AsteRxU in 2020 and 2021. The systematic difference between PolaRx5 and ROC2 was less than one percent (Haase et al., 2021) as an evaluation of the ARO measurement precision. The processing of raw RF data from the GSS6450 using open-loop tracking is ongoing and thus not presented in the study. The results from two GNSS/INS systems in 2018 and 2021 were only used to evaluate the
positioning results and were not implemented in the final ARO data processing. Given the unique opportunities for deploying multiple types of instrumentation through collaborations with other projects, an important conclusion is that the ease of use and flexibility of the PLL receivers, especially the Septentrio PolaRx5 and AsteRxU, provided the best solution for the current use case. The GNSS/INS systems did not provide enough of an advantage in accuracy at 1 Hz sampling to merit the additional complexity of operations nor the additional the cost of equipment and correction service. Although real-time GNSS/INS did
improve over standard autonomous real-time positioning, lower noise was achieved in the post-processing using precise point positioning (PPP) methods. Since AR Recon 2022, the raw ARO data (1 Hz sampling rate) have been transmitted via SATCOM in near-real-time and tests confirmed it could be processed on the ground with a $\sim$ 30 min delay. In summary, the stand-alone PLL GNSS receivers are the most cost-effective solution if the data are recovered post-flight or via SATCOM and processed on the ground. However, the balance of operability versus accuracy could be re-evaluated if on-board processing is eventually
considered.

## 2.4 ARO retrieval procedures

The algorithm for ARO refractivity retrieval is based on the work of Healy et al. (2002), developed by Xie et al. (2008), and initially implemented as described in Haase et al. (2014, Supplementary Information) and Murphy et al. (2015). Significant improvements were implemented in the latest version as described in Cao et al. (2022) for balloon-borne RO and briefly





summarized here. The main procedures for the ARO retrieval involve the following steps: (a) precise positioning, (b) phase residual calculation, (c) excess phase/Doppler conditioning, (d) bending angle calculation and (e) refractivity retrieval.

Precise point positioning (PPP) with ambiguity resolution (PPP-AR) (Geng et al., 2019) is used to calculate the precise positions of the aircraft. GNSS satellite orbit and clock products are required to implement the PPP calculation. The multi-GNSS satellite orbits, clocks, attitude quaternions, and earth rotation parameters (ERPs) are provided by the Center for Orbit

Determination in Europe (CODE) (Prange et al., 2020), and the GNSS Research Center of Wuhan University (WHU) (PRIDE Lab/Wuhan University, 2022b), under the Multi-GNSS EXperiment (MGEX). Currently, this procedure is implemented in post-processing mode after the final products are released after $\sim$ 14 days. The excess phase is the difference between the observed phase in the presence of the atmosphere and the calculated straight-line distance for the same transmitter-receiver geometry in a vacuum. This excess phase was calculated with aircraft positions fixed and satellite antenna phase center and relativity effect

removed. The ionospheric effort was eliminated by the linear combination of the phase of dual-frequency observations. Excess Doppler was then estimated by differentiation of excess phase with time. Receiver clock error was eliminated by a single-difference method in which the excess Doppler from a satellite at high elevation was subtracted from that of the occulting satellite. A second-order Savitzky-Golay filter was used to smooth high-frequency noise in the excess Doppler resulting from variabilities with a scale shorter than the first Fresnel zone. The filtering window width was determined to be 51 seconds based

on the vertical resolution analysis presented in the subsequent section.

In a spherically symmetric atmosphere, the ray path bending angle can be derived from the excess Doppler shift (Vorobev and Krasilnikova, 1994; Hajj et al., 2002; Kursinski et al., 1997) and known aircraft/satellite positions and velocities using an iterative method, geometric constraints, and Bouger's law for optical refraction (Born and Wolf, 1999). In order to use the approximation of spherical symmetry, the local radius of curvature, $R_c$, tangent to the Earth surface in the occultation plane is

calculated at the lowest tangent point location to correct for the oblateness of the Earth. The aircraft and satellite positions are shifted relative to the new center of curvature.

To reduce noise due to aircraft position uncertainty due to turbulence, the time series of positive bending angle and a portion of the negative bending angle from the flight level to 1 km below the flight level are smoothed with a 5-minute moving window. In the final optimized bending angle, a taper function is used to weight the transition from the raw to a smoothed bending

angle between 0.5 km and 1.0 km below the flight level. At each impact parameter, $a$, the partial bending angle, $\alpha'$, at each impact parameter, $a$, was calculated by subtracting the positive elevation angle observations from the negative elevation angle observations for the same impact parameter. The last step is to retrieve the refractive index from the bending angle using the Abel transform and then convert it to refractivity. The final meteorological parameters, currently dry pressure and temperature, are retrieved based on a simplified hydrostatic assumption. Since the main use of the data is for assimilation, the retrieval

of humidity using the variational method with model products as constraints is not currently carried out to avoid introducing errors from using model as a first guess.

The horizontal location (latitude/longitude) of a given tangent point height can not be determined explicitly due to the asymmetric geometry, thus it is determined by a forward simulation using ray-tracing in the Radio Occultation Simulator for Atmospheric Profiling (ROSAP) (Hoeg et al., 1996; Syndergaard, 1998) assuming a refractivity profile from the climatological





CIRA-Q model appropriate for the month and latitude (Kirchengast et al., 1999). The simulated tangent point locations are sufficiently close the actual ones, given they do not depend strongly on horizontal variations in refractivity, especially in the higher atmosphere. The final ARO profiles are described by a series of 4-D coordinates of the tangent points, including latitude, longitude, Mean Sea Level (MSL) altitude, and time for which refractivity, bending angle, dry pressure and try temperature are provided. The location and time of the lowest tangent point of the profile are chosen as the reference occultation point for the

ARO profile.

## 3  Results

### 3.1  AR Recon campaigns and ARO dataset

The objective of AR Recon is to collect supplemental observations in ARs and essential atmospheric structures to understand and improve forecasting of ARs and their precipitation impacts (Ralph et al., 2020). The flight planning for the G-IV and WC-

130s is guided by calculations of forecast sensitivity to observations using adjoint (Doyle et al., 2014; Reynolds et al., 2019) and statistical sensitivity methods (Torn and Hakim, 2008; Ancell and Hakim, 2007). Over the four-years of AR Recon campaigns from 2018–2021, a total of 33 intensive observation period (IOP) flights were carried out by the NOAA G-IV (Table 2). In 2019 and 2021, there were an additional 3 and 4 ferry flights of the G-IV between Hawaii and the continental US. For some IOPs, one or two USAF WC-130s joined the reconnaissance missions to perform coordinated observations. There were also some

IOP flights executed solely by WC-130s when the G-IV did not participate due to aircraft and crew availability. Experimental ARO equipment was deployed on the WC-130s in 2020 and 2021. However, at the time, the WC-130s were equipped with L1-only GPS antennas. This makes it difficult to eliminate ionospheric effects, which is required for precise positioning and phase residue retrievals. The data were archived for those years, pending the development of suitable alternative approaches.

In 2018 and 2020, the G-IV aircraft was based in Seattle and Portland, respectively, from which the aircraft flew out over

the northeastern Pacific where the ARs can be observed prior to landfall on the west coast. The flights cover a large area with a latitudinal range from southern Alaska to almost Baja California. In 2019, the G-IV flew multiple GRAV-D gravity surveys for the NOAA National Geodetic Survey (NGS) near Hawaii and Samoa, around the same time as the AR Recon missions. This provided multiple ferry flights and measurements of opportunity between the continental US and Hawaii. We processed the data from three ferry flights between California and Hawaii, where the aircraft flew over the area of greatest AR influence.

In 2021, the operational base was relocated to Honolulu, Hawaii, from which the aircraft could fly over a much broader area of the North Pacific, complementing the WC-130 aircraft based on the US west coast. With the growth of AR Recon, additional resources permitted multiple consecutive (back-to-back) flights over the span of the genesis and evolution of the same synoptic system. Observations suggested a greater impact on the forecasts from a sequence of flights (Zheng et al., 2021) than individual flights, in the period leading up to the precipitation associated with the landfall of the ARs. The approach for longer sequences

of daily sampling of ARs was adopted in 2021. In 2021, we also processed and included in the dataset the four trans-Pacific ferry flights between California/Arizona and Hawaii. Table 2 lists the start/end dates for G-IV participation in the campaigns and the number of IOP and trans-Pacific ferry flights each year.





**Table 2.** Number of flights over the four years of AR Recon campaigns.

| Year | Start date[a] | End date[a] | Operation base | # IOPs | # trans-Pacific ferry |
|------|------------|----------|----------------|--------|----------------------|
| 2018 | 01/26 | 02/03 | Seattle, WA | 3 | 0 |
| 2019 | 02/01 | 03/15 | | 0 | 3 |
| 2020 | 01/24 | 02/25 | Portland, OR | 13 | 0 |
| 2021 | 01/15 | 02/26 | Honolulu, HI | 17 | 4 |

[a] The start and end dates are solely for the availability of the ARO dataset from the G-IV as the campaign might start earlier and last longer for flights executed by the USAF WC-130s.

**Table 3.** Numbers of ARO profiles each year, with non-IOP trans-Pacific ferry flights included.

| Year | GPS | GLONASS | Galileo | Rising | Setting | Total | Flight hours[a] | # per hour | Interval[b] |
|------|-----|---------|---------|--------|---------|-------|-------------|-----------|-----------|
| 2018 | 75 | 0 | 39 | 49 | 65 | 114 | 22.0 | 5.2 | 11.9 (17.3)[c] |
| 2019 | 27 | 16 | 19 | 29 | 33 | 62 | 9.1 | 5.7 | 11.4 (21.6) |
| 2020 | 335 | 154 | 197 | 325 | 361 | 686 | 93.9 | 7.3 | 7.8 (17.6) |
| 2021 | 374 | 255 | 243 | 364 | 508 | 872 | 141.1 | 6.2 | 8.9 (20) |
| Total | 811 | 425 | 498 | 767 | 967 | 1734 | 266.1 | | |

[a] Flight hours include only flight segments where the aircraft flew above 9 km.

[b] The unit of "interval" is minutes on average between occultations.

[c] The numbers in the parentheses are intervals that count only GPS occultations.

Table 3 lists the number of ARO profiles retrieved from different constellations (GPS, GLONASS, and Galileo), their type (setting or rising), and the total number of flight hours in each mission year. Valid ARO profiles are retrieved after the aircraft reaches cruise altitude. The flight hours in Table 3 only count the flight segments above 9 km, which exclude aircraft ascent and descent, and are, therefore, about half an hour less than the total duration of each flight. There are 1734 profiles retrieved over the four mission years from the three major constellations, with ∼ 25% more setting occultations than rising ones. There are 5–7 profiles retrieved per hour, corresponding to an average sampling interval of 8–12 min. Implementing multi-GNSS observations more than doubles the number of profiles and reduces the interval by half, compared to GPS-only observations. In 2018, the average interval was reduced from 17 min to 12 min when the Galileo constellation was added. Occultations from BeiDou constellations are currently being evaluated, which would reduce the average interval further. Figure 3 shows the distribution of the ARO profiles over the flight duration for all flights. The profiles are not evenly distributed over time but roughly cover the whole flight. The variation in the number of profiles per flight does not directly reflect the performance of the ARO observations, because the maximum number of occultations of each flight depends on the flight track, orientation, and timing relative to the visibility of the GNSS satellites. The flight time was planned such that dropsondes were released within the ±3-hour window around 00:00 UT over the target of interest for the desired day for data assimilation. About 80% of the profiles (1360 out of 1734) are within the 6-hour target DA window, with the remaining 20% spread over the ferry portion of





the flights. The 8–9 min average interval is very close to the 10 min minimum interval between dropsondes and corresponds to about 100 km separation at typical flight speeds between two profiles at the highest points.

**Figure 3.** The retrieved ARO profiles are organized by flight time (UT) and flights. Upward and downward triangles indicate rising and setting occultations, respectively. Red, blue, and black denote occultations retrieved from GPS, GLONASS, and Galileo constellations. The two numbers in the parentheses are the number of ARO profiles within the DA window (6-hr centered at 00:00 UT) and the total number for that flight. The date "year.doy" (day of year) is the aircraft take-off date, corresponding to the date before 00:00 UT, and the information of "flight-id' can be found at https://cw3e.ucsd.edu/arrecon_data/.





Figure 4 shows all retrieved ARO profiles for each mission year, including IOP and trans-Pacific ferry flights. The signal ray paths that connect the satellite transmitter and the aircraft receiver traverse the atmosphere from flight level to the surface during a setting occultation (or vice versa for a rising one) to form a complete profile. Because the aircraft flies much slower than the GNSS satellites ($200 \, \mathrm{m \, s^{-1}}$ vs. $3900 \, \mathrm{m \, s^{-1}}$) the tangent point of the ray path drifts horizontally away from the aircraft as each subsequent ray path traverses the atmosphere vertically (Fig. 5b). When the GNSS satellite signal arrives exactly

horizontally relative to the aircraft, the tangent point is at the aircraft position. The slant of ARO profiles is generally larger than space-borne RO profiles with same altitude range. In map view (Fig. 4), the slanted ARO profiles appear as curves starting from a point on the flight track and ending approximately 400–600 km away from the flight track. The red and blue curves indicate setting and rising occultations, respectively. While the dropsondes observe the area directly underneath the flight track, the ARO observations expand the aircraft sampling to a broader area, observing the gaps between and around the flight tracks.

As shown in Fig. 3 and Fig. 4, retrieved ARO profiles are irregularly but densely distributed over the flight time and track. The patterns of the tangent point drift, azimuthal dependence, and duration of ARO observations are analyzed in the following sections.

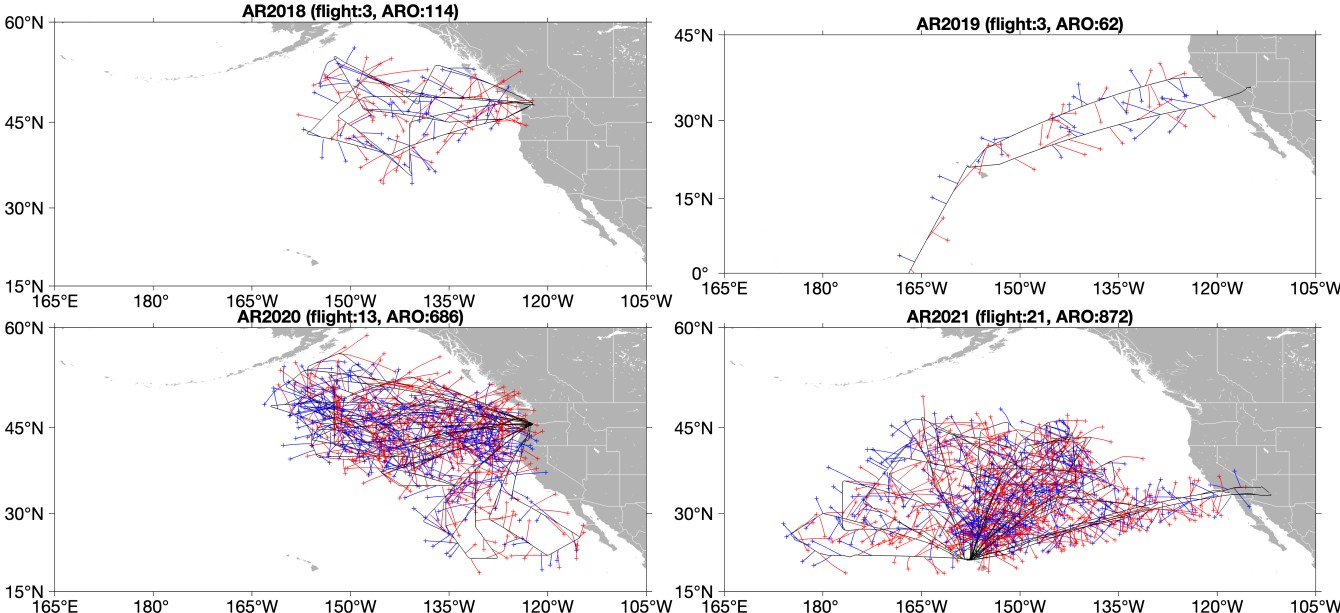

**Figure 4.** Maps of the horizontal projection of ARO profiles from four-years of AR Recon missions. Black lines indicate the flight tracks and red and blue lines denote setting and rising occultations, respectively. Occultations from different constellations are not distinguished in the maps. The small crosses at the end of the profiles mark the lowest tangent point location, which is the reference point given in the metadata.



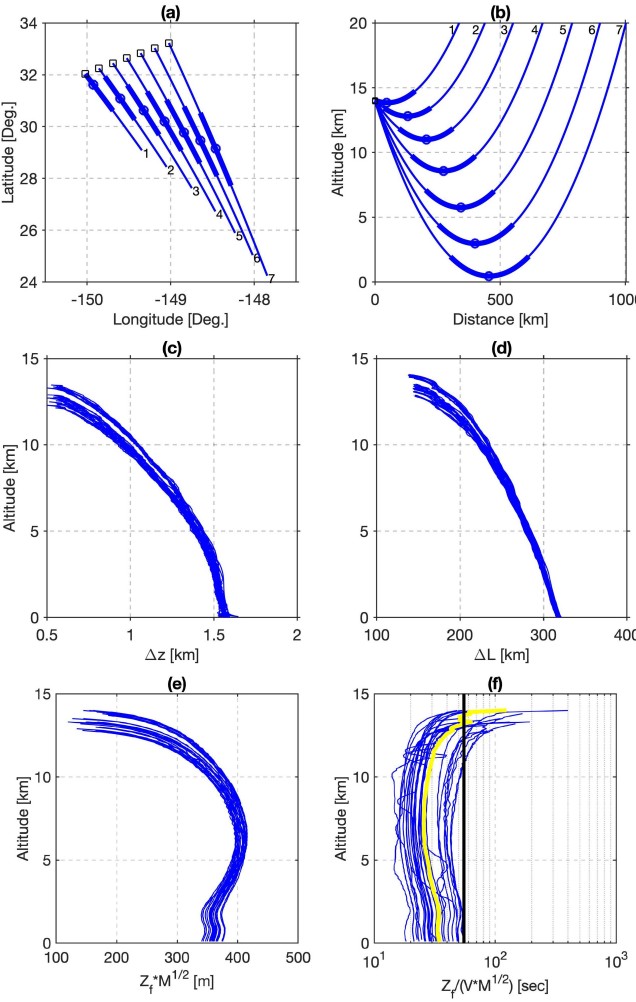

**Figure 5.** Shift of the ray paths for each tangent point of one rising occultation ("g02r" of IOP06 in AR Recon 2021) in (a) horizontal plan view and (b) vertical cross-section along the ray path orientation with the ray path truncated at 20 km. The numbers 1 to 7 label the ray paths from top to bottom in 125-sec intervals. Black squares mark the aircraft's position as it flew toward the southwest. Blue dots on each ray path mark the tangent point. The thick segment surrounding the tangent points indicates the path length over which 50% of the excess phase is accumulated. (c) The vertical distance and (d) the corresponding horizontal distance separating segments that contribute 50% of the excess phase along the ray paths. (e) The diameter of the first Fresnel zone, including the effects of defocusing in the atmosphere, and (f) the time required for the ray path to scan the vertical distance equivalent to the diameter of the first Fresnel zone. The blue lines in (c)–(f) are for all GPS occultations retrieved in IOP06 of AR Recon 2021. The thick yellow line in (f) is the mean value, and the black vertical line denotes 55 seconds.





## 3.2 Horizontal and vertical resolution

RO is a limb-sounding technique with the advantage of high vertical resolution, especially in the lower troposphere where it
is better than 1 km (Zeng et al., 2012). The observed refractive delay and bending used to derive the refractivity are integrals
over the length of the ray path, so it is an approximation to treat the derived refractivity as a local measurement at the tangent
point. There are multiple perspectives to define the vertical/horizontal resolution. One way is to presume that the sampling
region of an individual independent observation corresponds to the length of the ray path surrounding the tangent point that
contributes 50% of the excess phase or bending. This would lead to an approximate resolution defined by the vertical and
horizontal spatial range separating two such observations. Figure 5(a) illustrates the signal ray path traversing the atmosphere
during one rising occultation event (labeled as "g02r") in horizontal plan view and Fig. 5(b) shows the vertical cross-section
as a function of distance from the aircraft. The ARO receiver continuously tracked a GPS satellite (PRN# 02) as it rose from
a negative elevation to high above the horizon. The blue lines illustrate the signal ray paths originating from the GPS satellite
orbiting at a much higher altitude (not shown in the figures) and ending at the aircraft, which is flying toward the southwest.
Only segments of ray paths below 20 km are shown in the figure, and the interval between two adjacent ray paths is decimated
to 125 sec from the raw 1-sec sample interval, to illustrate better the horizontal drift of the tangent points. Although the actual
ray paths are typically bent downwards (with $\sim 1°$ bending angle) relative to an ellipsoidal Earth, the ray paths are shown here
curved upwards relative to the flattened Earth surface. The thicker segments on the ray paths near the tangent points account
for 50% of the total accumulated excess phase. The altitude limits, $\Delta z$, of this segment and corresponding horizontal distance,
$\Delta L$, provide one way of defining the resolution. Because the aircraft changes course and different GNSS satellites orbit in
different directions relative to the aircraft, the occultation geometry as shown in Fig. 5(a) and 5(b) varies for different ARO
profiles. We estimated $\Delta z$ (Fig. 5(c)) and $\Delta L$ (Fig. 5(d)) for all GPS occultations retrieved for one flight (IOP06 of AR Recon
2021). Despite the different occultation geometries, $\Delta z$ and $\Delta L$ are very similar among different occultations and vary from
1.5 km and 300 km near the Earth's surface to 0.5 km and 150 km at flight level, respectively.
Theoretically, the resolution is based on the diffraction of the ray path and is defined as the diameter of the first Fresnel
zone (see detailed definitions and formulas in Kursinski et al., 1997; Haase et al., 2021; Cao et al., 2022). Figure 5(e) shows
the diameter of the first Fresnel zone ($Z_f$) with atmospheric defocusing effects ($M$) considered and Figure 5(f) shows the
time it takes for ray paths to scan the altitude range equivalent to that diameter. The resolution represented by the diameter
is about 400 m between 5–10 km, decreasing near the surface to $\sim 350$ m due to the atmospheric defocusing in stronger
near-surface refractivity gradients. The resolution reduces to $\sim 200$ m at flight level due to the tangent point being close to
the aircraft/receiver. The ray paths require an average of 30–50 seconds to scan the distance of the first Fresnel zone diameter,
which determines the window width used for filtering in the retrieval procedures that would smooth out any fluctuation with
scales less than that diameter. The 1-sec sample interval of the data is well below that requirement and would correspond to a
distance of roughly 10–20 m between the two closest ray paths under the geometric optics assumption in the absence of noise.
As discussed above, the greatest contribution to the observed refractivity at the tangent point derives from an integral over a
horizontal distance of 300 km along the ray path. However, the resolution perpendicular to the occultation plane is also defined



by the diameter of the first Fresnel zone and, therefore, is as high as the vertical resolution. ARO profiles can resolve small-scale variations in the vertical direction and in the direction perpendicular to the occultation plane as the tangent point drifts horizontally as shown in Fig. 5, but not in the direction along the ray path. The variations in the vertical direction are accounted

for in the retrieval and also in the simulation of the observations from numerical weather prediction (NWP) model fields for the purpose of data assimilation by using a 1-D observation operator for refractivity or bending angle. However, this might not be the case for horizontal variations in the atmospheric properties, for example, when encountering strong horizontal refractivity gradients due to fronts or associated ARs (Xie et al., 2008). The orientation of the ray paths and the tangent point drift direction relative to the sensing targets matter and should be considered when interpreting profiles and assimilating into NWP models.

The use of bending angle rather than retrieved refractivity in the assimilation reduces the impact of this sensitivity in the retrieval process and reduces error correlations. Using a 2-D operator that integrates model properties along the ray path rather than a 1-D operator may be necessary to accommodate the effect of horizontal variations in the atmosphere (Eyre, 1994; Chen et al., 2018; Hordyniec et al., 2024; Murphy et al., 2024).

## 3.3 Spatial distribution

The timing and location of the ARO profiles have a quasi-random pattern along the flight tracks. Figure 6 shows an example of the spatial distribution of all ARO profiles of one flight (IOP04 of AR Recon 2020 centered on 00 UT Feb 4, 2020). The clockwise flight track (thin black line) was designed to transect the AR, as represented by the magnitude of vertically Integrated Water Vapor Transport (IVT). The IVT is estimated by integrating specific humidity multiplied by the wind for a vertical column of the troposphere (surface to 300 hPa), and represents the horizontal transport (flux) of moisture in the

atmosphere. The width of the AR reaches 800–1000 km, using an IVT threshold of 250 $\mathrm{kg\,m^{-1}\,s^{-1}}$ as the limits. This AR event was associated with an ETC in the Gulf of Alaska and stretched almost 4000 km across the Pacific from Hawaii to Canada, making landfall on 06 UT on 5 February with heavy precipitation in the states of Oregon and southern Washington. At 00 UT on 4 February, the magnitude of IVT reached 1000 $\mathrm{kg\,m^{-1}\,s^{-1}}$ in the core of the AR. There were 30 dropsondes released (circles in Fig. 6), creating two straight-line transects across the AR core. 56 ARO profiles were retrieved and extend

the sensing area to more than 500 km around the flight track. The penetration depth of the profiles varies, with at least 7 of them reaching below 3 km, and 12 more between 3 and 6 km. Several profiles extend outside the flight track upstream and downstream along the AR core with deep penetration. On the outbound and inbound segments of the flight east of the AR, no dropsondes were released, but 21 ARO profiles were retrieved, illustrating how ARO can sample in critical near-shore areas where dropsonde releases are controlled or prohibited. In Fig. 6, the GNSS signal ray path orientation is shown by short lines

indicating the length over which 50% of the total excess phase is accumulated. The signal ray path of each occultation is typically oriented subparallel to the tangent point drift direction, within 20°–30°. Only the orientation of the lowest ray path is shown, but the orientation at higher altitudes is approximately the same (Fig. 5(a)). The typical horizontal integration length ($\sim$ 300 km) is much shorter than the size of the synoptic scale AR feature. For a closed-circuit flight track such as this, multiple occultations penetrate into the center of the circuit from different directions. This tomographic-style scanning of the target area





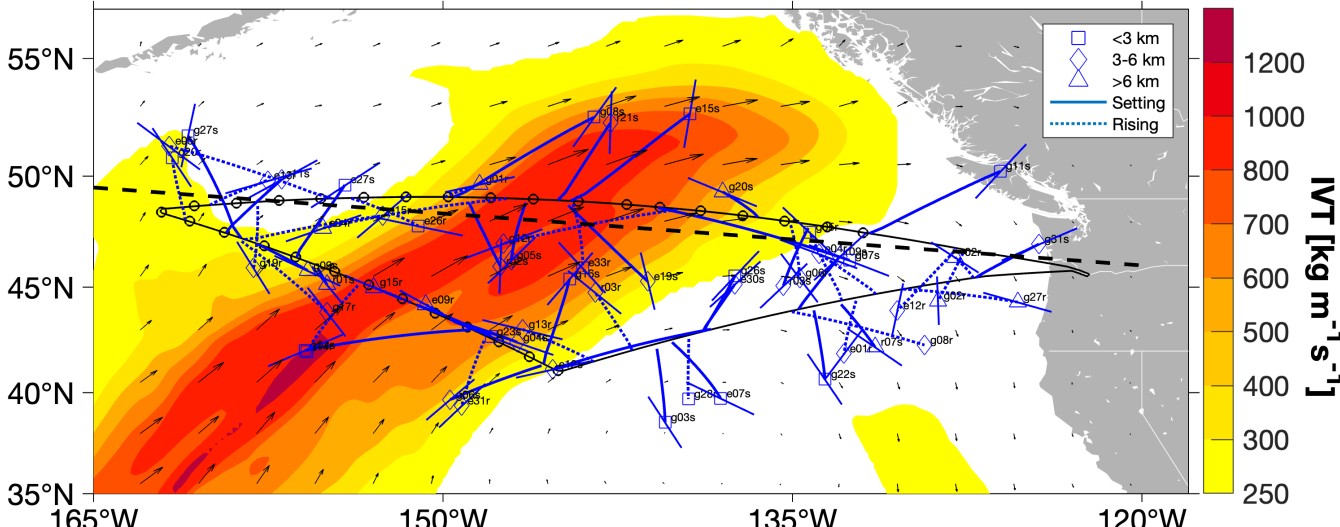

**Figure 6.** Distribution of ARO profiles from IOP04 in AR Recon 2020 centered on 00 UT Feb 4, 2020. The thin black line indicates the clockwise flight track. Black circles indicate the position where dropsondes were released. The thick solid and dotted blue lines denote the projection of tangent points location for setting and rising occultations, respectively. Occultations are labeled with GNSS satellite number (PRN) and type (setting/rising), prefixed with 'g' for GPS, 'r' for GLONASS, 'e' for Galileo, and suffixed with 'r' for rising and 's' for setting. The symbols at the end of the lines mark the lowest point with squares, diamonds, and circles indicating the lowest point below 3 km, 3–6 km, and above 6 km, respectively. The thinner short lines around the lowest point denote the segment of the ray path at that azimuth contributing to 50% of the excess phase, as defined in Fig. 5b. The shaded contours are the magnitude of vertically Integrated Water Vapor Transport (IVT) with arrows showing the magnitude and direction of transport, see text for definition. The dashed black line denotes the location of a transect that is closely aligned with the last flight segment and is shown in Fig. 11.

combined with the dropsondes provides dense coverage of the AR core. This is an attractive property that could be exploited
for hurricane reconnaissance, for example, for flights designed to circumnavigate the targets.

 Given the planned aircraft trajectory and forecast GNSS satellite orbital ephemerides, one would expect that the timing and
location of ARO profiles can be predicted beforehand to achieve a specific ARO distribution pattern. However, a slight change
in flight timing, such as a delayed take-off, would lead to a very different spatio-temporal distribution of ARO profiles. The high
sensitivity to timing makes it impractical to utilize this feature to design a flight track. Below, we analyze the general pattern of
horizontal drift and azimuthal dependence so that flight tracks can be designed to take full advantage of the expanded sensing
area within the AR and the surrounding environment.

### 3.4 Profile obliqueness

The horizontal drift distances for all occultations in the dataset are shown in Fig. 7, relative to the location and height of the
highest point in the profile. The highest point matches the aircraft altitude, which typically varies between 13–14 km altitude
for all profiles from the G-IV. This drift distance depends on the geometry of the occultation plane, given by the positions and



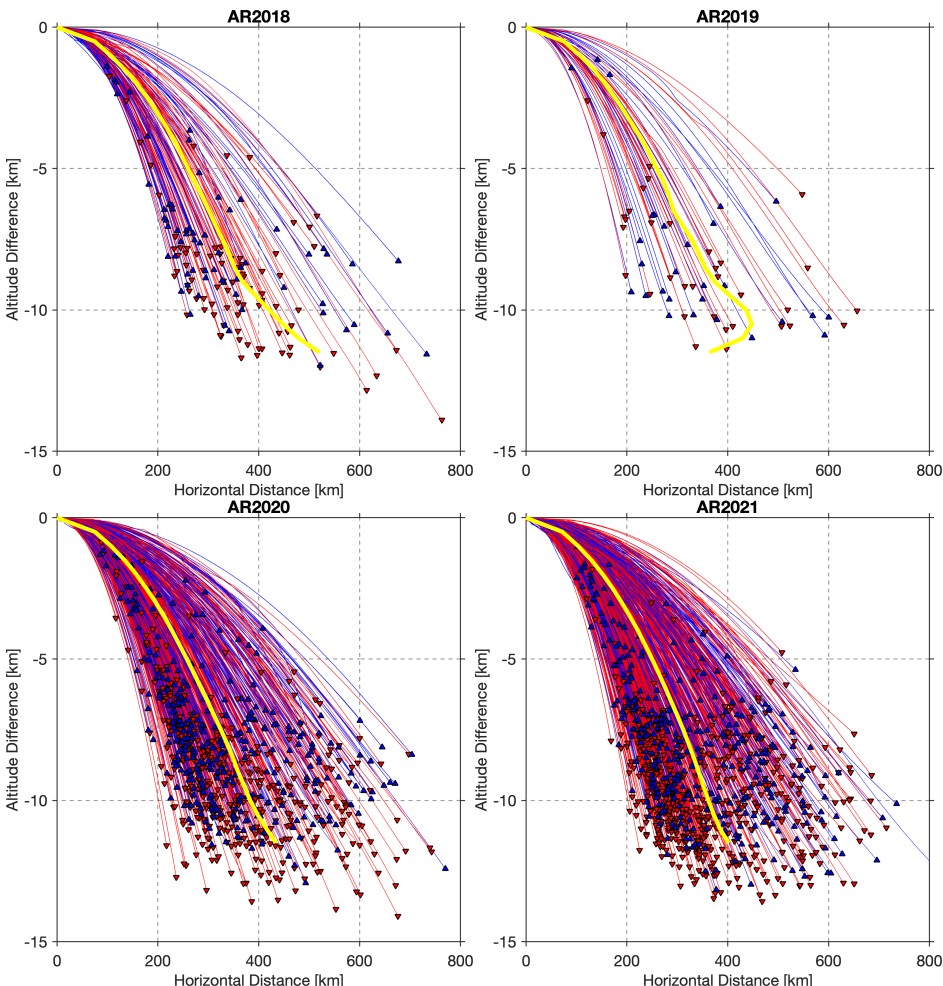

**Figure 7.** Horizontal drift as a function of height below flight level. Red and blue lines denote the setting and rising occultations, respectively. The thick yellow lines are the average over all profiles of drift distance as a function of height below flight level. The triangles at the end of each curve mark the lowest and furthest point, with upward and downward triangles indicating rising and setting occultations, respectively.

velocities of the aircraft and satellite. However, the final distance of tangent point drift and duration is more strongly controlled by the success of the tracking through the tropospheric structure rather than simply the geometry. The data collected over each of the four years show similar ranges of drift distance. The average horizontal drift is about 400 km and can be as much as 700 km when the penetration depth is $\sim 13$ km, and the lowest point approaches 1–2 km MSL altitude. The horizontal drift rate is greater at the top of the profile, then gradually reduces to become more linear toward lower altitudes. There is a $\sim 200/8$ horizontal-to-vertical drift ratio for tangent points more than 2 km below flight level. The obliqueness makes direct interpretation of the profiles complex, compared to near-vertical profiles such as dropsondes and radiosondes. However, it is this horizontal drift that provides the unique advantage of extending the sensing area from directly beneath the flight track

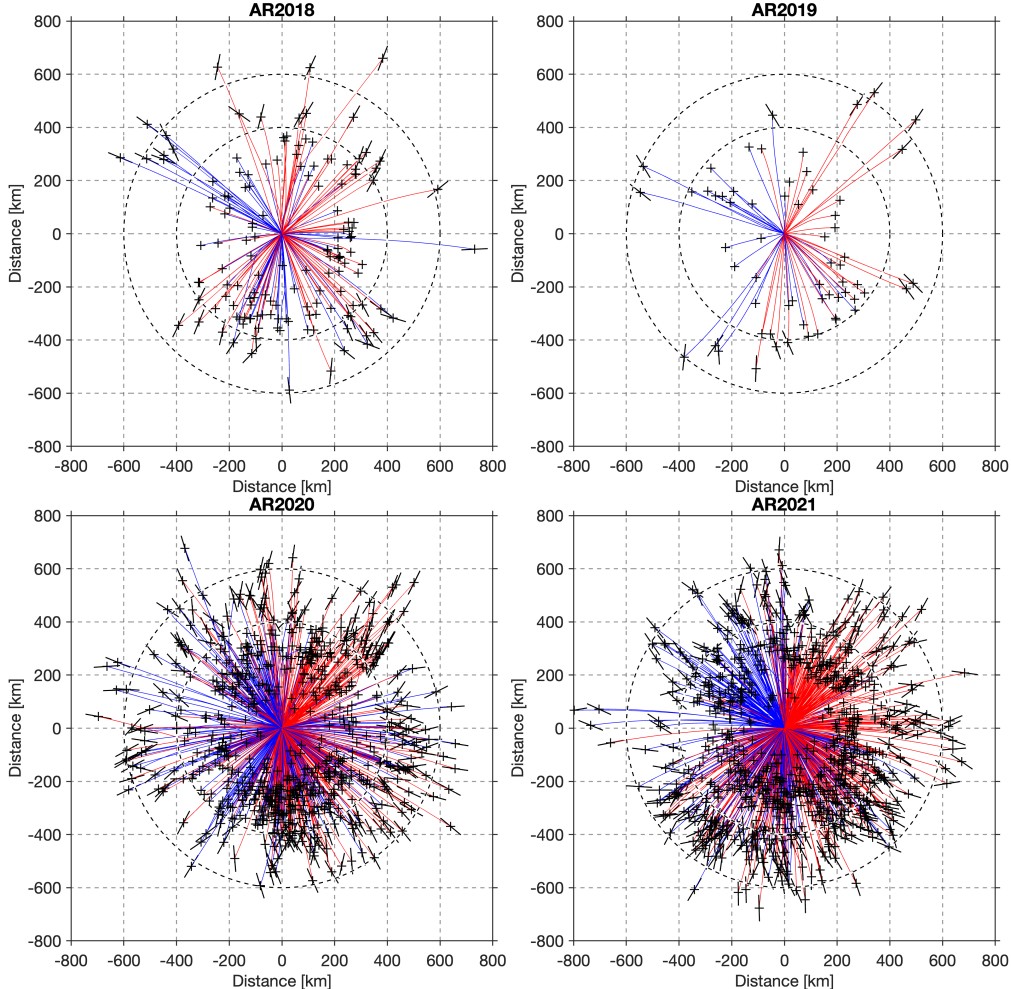

**Figure 8.** Plan view of the horizontal drift and orientation of ARO profiles, relative to the position of the highest point of each profile, with red and blue denoting setting and rising occultations, respectively. The crosses at the end of each curve mark the lowest and furthest points. The short thick lines around the crosses denote the signal ray path orientation at that lowest altitude. The signal ray path orientation is shown only for profiles with drift distances more than 400 km. There is a 20–30° difference between the signal ray path orientation and the tangent point drift.

to a wider band $\sim$ 400 km to both sides of the track. Combining the simultaneous ARO and dropsonde observations makes it possible to resolve variations in the horizontal structure of the AR perpendicular to the flight track, as demonstrated for horizontal temperature gradients in Haase et al. (2021).

The tangent point drift projected onto the horizontal plane, centered at the location of the lowest point, shows the azimuthal distribution of the slant profiles (Fig. 8). The orientation of the signal ray path (occultation plane) is marked by short lines at

the end of the profile, usually within 20°–30° of the tangent point drift direction. The profiles retrieved from the four years of





missions show a preferential direction toward the northeast for setting occultations and the northwest for rising ones. This is most clear for the ferry flights from the West Coast to Hawaii in 2019 and 2021, where the flight paths are roughly east-west (Fig. 4). This anisotropic pattern in azimuth is related to the fact that the GNSS satellite orbital inclination angles are 55°–65°. At the equator, the orientation of tangent point drift is bimodally distributed NE-SW or SE-NW (i.e., see Cao et al., 2022) because the GNSS satellites tend to set or rise at those azimuths. The ferry flights at low latitudes reproduce this effect (Fig. 8(b)). However, over the four years of AR Recon missions, the aircraft flew over a large latitudinal range (20–50° N) at various headings. When in the northern latitudes, the orientation will still preferentially lie in the NE and NW directions for rising and setting occultations that occur on the north side of the aircraft. However, on the south side of the aircraft, satellites will also be visible over a range of southerly directions, producing a broader and more random distribution of orientations. The pattern would be reversed if there were flights in the southern latitudes. The inclined GNSS satellite orbital planes result in some azimuths with fewer ARO profiles retrieved, particularly in the east-west direction and close to due north.

### 3.5 Duration and penetration depth

The duration of one occultation can be defined as the difference between the time when the signal from the satellite arrived directly horizontally as viewed from the aircraft (highest point of the profile) and the time the receiver lost or initiated signal tracking (lowest point of the profile). In general, the occultations will be shorter in duration, and the tangent point will drift a shorter distance when the aircraft velocity has a component in the anti-velocity direction of the GNSS satellite, i.e. when the aircraft flies towards a rising GNSS satellite or away from a setting GNSS satellite. For ARO, the duration of the occultation is controlled by the speed the GNSS satellite sets, as opposed to the speed the LEO sets for SRO. Therefore, the typical ARO duration is much longer than SRO ($\sim$ 100–200 sec). However, the overall controlling factor for the duration is the penetration depth rather than geometry. The duration and penetration depth of ARO profiles for each year are shown in the scatter plots in Fig. 9, along with their corresponding histograms. The duration lies in the range of 5–15 min, with an average of about 10–11 min. This duration is sufficiently short to resolve most synoptic scale atmospheric variations. The average penetration depth is about 8–9 km below the aircraft flight level, which for the typical 13–14 km G-IV flight altitude corresponds to the lowest height at around 4–6 km MSL altitude. The occultations which penetrate to lower altitudes generally have a longer duration. Results from all four years show a very similar distribution with slightly different mean values. The relationship between penetration depth and duration is quasi-linear for profiles shorter than 10 min, with a descent rate of $\sim$ 1 km min$^{-1}$. However, some occultations last longer than 20 min without penetrating any deeper, due to a geometry where the GNSS satellite sets or rises sideways rather than perpendicular to the horizon. For 2021, the average duration was about one minute longer, and the penetration depth was about 0.5 km deeper than in previous years. This is due to the upgrade of the GPS-only to a multi-GNSS antenna whose broader frequency band and higher pre-amplifier gain enabled better signal tracking, especially for GLONASS and Galileo satellites.

Figure 10 shows the cumulative probability distribution of the lowest measurement altitude for occultations of different constellations and types. This lower limit is determined by multiple factors, including but not limited to receiver/antenna performance, atmospheric conditions, possible obstruction of the signal, and quality control in the data processing methods





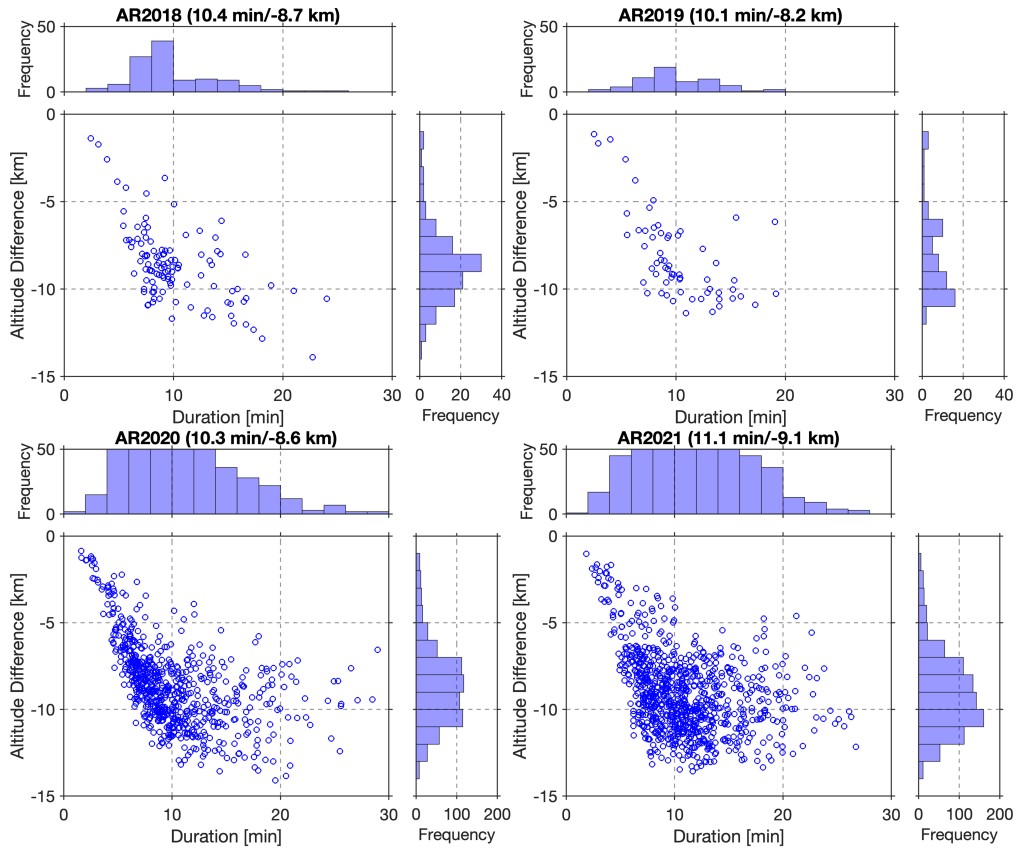

**Figure 9.** Scatter plots of the ARO profile duration with respect to the penetration depth of the lowest tangent point altitude below flight level. The histogram for the duration is shown at the top, and the histogram for the penetration depth is shown to the side. The numbers in parentheses are the mean duration and penetration depth for each campaign year. Flight level averages ∼ 13–14 km for the G-IV.

(i.e., the threshold for excluding data with gaps). There is a sharp drop in the number of observations in the range between 2–3 km, most likely due to strong gradients near the top of the boundary layer. These can produce significant fluctuations in phase and amplitude due to atmospheric multipath that limit the performance of the PLL tracking receivers. In the four years of AR Recon missions, most profiles penetrated below 5 km, with an average lowest height of around 4.4 km. Comparing the occultations of different types, the setting occultations tend to penetrate ∼ 1.4 km lower on average than the rising occultations.

The most significant improvement, however, was for GLONASS satellites between 2020 and 2021 when the antenna was upgraded to multi-GNSS, providing a broader bandwidth and higher gain. GLONASS signals were tracked much lower, so that the median lowest altitudes dropped by 1.3 and 1.5 km for setting and rising occultations in 2021 compared to 2020. It is not clear why there is a slight decrease in performance for GPS and Galileo in 2021. The wider-bandwidth antenna would account for the improvement of setting occultations, however, for rising occultations, it might be that more out-of-band noise led to 475 lower SNR for the GPS and Galileo constellations and thus fewer occultations. There is some variation among years that could





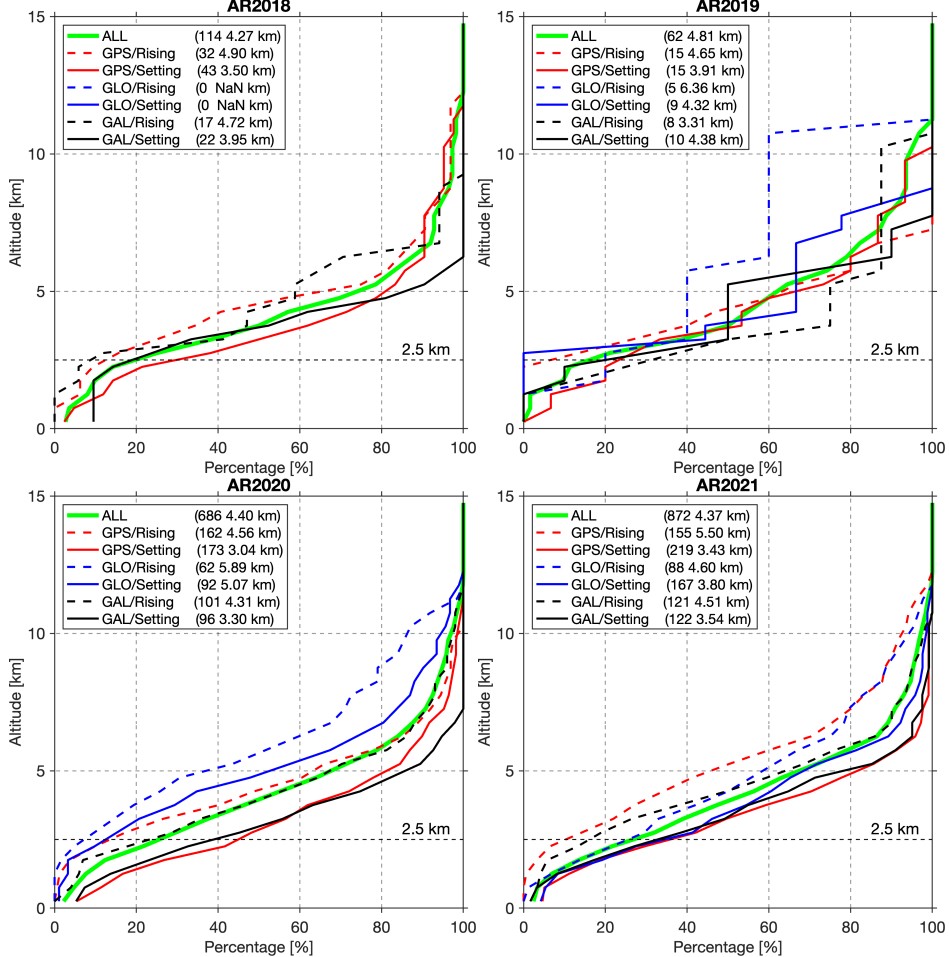

**Figure 10.** Cumulative percentage of occultations with lowest tangent point reaching below the given altitude (on y-axis) for different constellations as indicated by color. Rising occultations are indicated by solid lines and setting occultations with dashed. The numbers of profiles in each category and the median lowest altitudes are given in parentheses. The altitude of 2.5 km is marked for reference.

be attributed to the difference in the environment, where 2018 and 2020 had the most flights over the NE Pacific, and 2021 had most flights at more subtropical latitudes. In general, this is consistent with the observation that COSMIC-1 and other SRO profiles do not penetrate as low in the moist tropical atmosphere (Ao et al., 2012).

To illustrate the ARO sampling relative to the underlying AR environment, a vertical transect was created from the ERA-5
reanalysis that is closely aligned with the nearly straight northern flight segment (dashed line in Fig. 6) of IOP04 in AR Recon 2020. The locations of slanted ARO profiles recovered on this segment were projected onto the transect in local Cartesian coordinates and plotted as a function of longitude and MSL altitude (Fig. 11). Pressure, winds, temperature, and humidity were interpolated in 2-D along the transect from the ERA-5 reanalysis, and refractivity was also calculated from the model variables. The refractivity anomalies, defined as the difference from the mean refractivity profile of the whole transect, were calculated





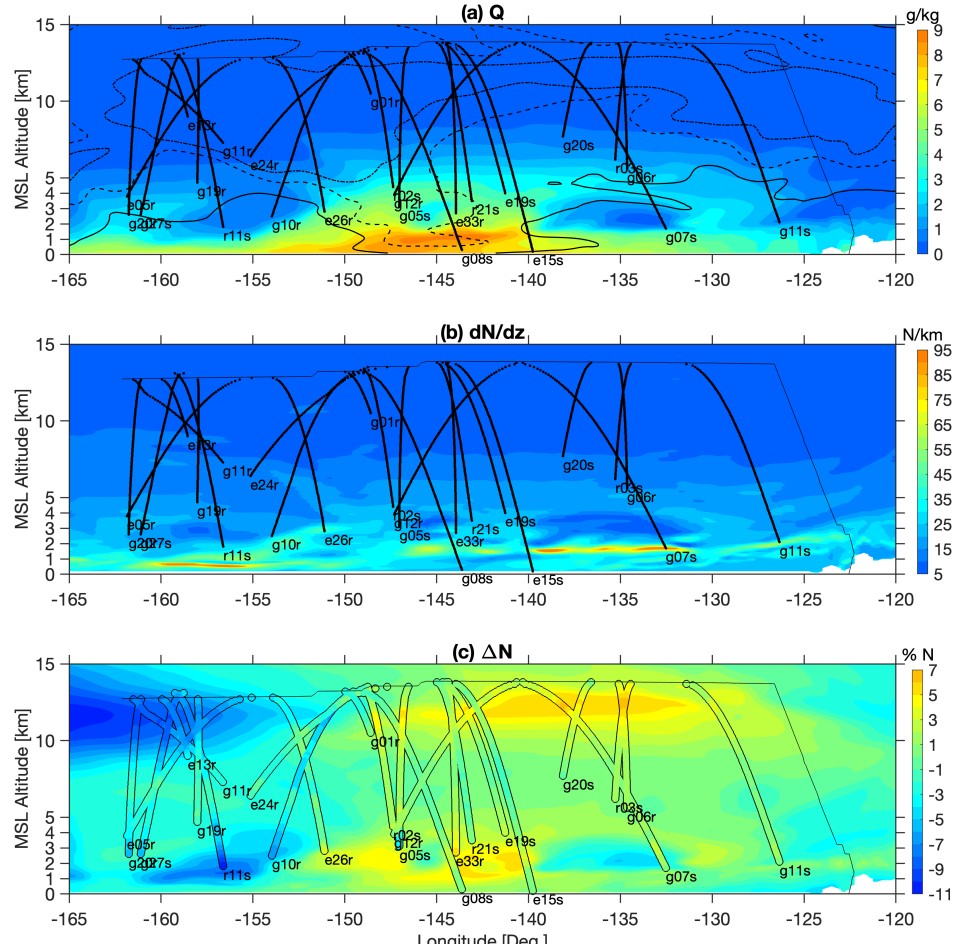

**Figure 11.** Vertical transect along the dashed line shown in Fig. 6 from IOP04 centered on 00 UT Feb 4, 2020, displayed as a function of longitude and MSL altitude. Color-shaded contours are (a) specific humidity, (b) refractivity gradient, and (c) refractivity anomaly in percentage differences interpolated from ERA-5 reanalysis. In panel (a), the solid, dashed and dotted contour lines denote wind velocity of $20 \, \mathrm{m \, s^{-1}}$, $30 \, \mathrm{m \, s^{-1}}$, and $40 \, \mathrm{m \, s^{-1}}$. In panels (a) and (b), the thick black lines, composed of a series of dots, are tangent point projections of the ARO profiles onto the transect. In panel (c), the refractivity anomalies are relative to the mean refractivity profile of the whole transect. The same refractivity anomalies are calculated for each ARO profile and shown by colored dots encircled by thin black lines to distinguish with the color shading in the background. The labels near the lowest point of each profile have the same meaning as in Fig. 6. The thin black lines near the top of each panel are the projection of the aircraft trajectory of the last flight segment.

and shown in Fig. 11(c). Although the transect is not perpendicular to the AR, the structure of the AR and its core can be seen in the specific humidity in Fig. 11(a). The high moisture in the core spanning from 150° W to 140° W is concentrated below 1.5 km, and values as high as $3.5 \, \mathrm{g \, kg^{-1}}$ extend to 5 km altitude. There are strong east-northeastward winds above and extending down into the AR core, leading to strong IVT toward the west coast of British Columbia (Fig. 6). Although the transect is not





perpendicular to the AR and jets, the moist low-level jet is evident in the 30 m s$^{-1}$ contour at 500–1000 m around 145° W
(Fig. 11). The observed value of refractivity anomaly (difference between the observation and the mean profile of the whole
transect) is plotted with the same color scale as the reanalysis at the location of each tangent point projected onto the transect
(Fig. 11(c)). Although the slanted ARO profiles do not sense the area directly beneath the flight track or in the plane of the
transect, the pattern of the ARO observed refractivity closely matches the ERA-5. This similarity reveals the capability of ARO
to resolve the AR structure and its synoptic environment (Haase et al., 2021; Murphy and Haase, 2022). Some differences
are seen at low levels for profiles such as 'g08s', 'r21s', and 'e15s' that stretch far to the side of the flight track. These three
subparallel profiles all probe the AR core downstream from the flight track, with the lowest tangent point sampling regions
of 800 kg m$^{-1}$ s$^{-1}$ as opposed to 1000 kg m$^{-1}$ s$^{-1}$ beneath the flight track (Fig. 6), thus explaining the differences in terms
of spatial variation perpendicular to the transect. Two profiles ('g08s' and 'e15s') reach the surface within the AR core, while
many other profiles terminate at an altitude where that is roughly coincident with refractivity gradients exceeding about 50
N km$^{-1}$ (Fig. 11(b)). The two deep profiles ('g08s' and 'e15s') appear to reach the surface through the gaps in layers with
sharp gradients. This highlights the advantage of collecting ARO profiles in addition to dropsondes, with its ability to sense a
wider environment beyond the flight tracks.

The G-IV missions often sampled the upper-level trough above the tropopause and in regions where the sensitivity of the
forecasted precipitation to potential vorticity errors was high (Reynolds et al., 2019). Above ∼ 9 km, the transect (Fig. 11)
shows variations in refractivity that are due to temperature rather than moisture (see Eq. 1). The ARO profiles at this level
are consistent with the ERA-5 variations. The high positive refractivity anomaly on the right side of the panel is where the
tropopause is higher than average (colder temperatures at 12 km) and the low refractivity anomaly on the left side of the panel
is where the the tropopause is lower than average (higher temperatures at 12 km). Several ARO profiles are high enough to
capture this change in lapse rate. The mid to upper troposphere ARO measurements are most reliable in terms of retrieval
accuracy, and provide valuable information on upper-level dynamics.

## 3.6 Obstruction of the signal

The GNSS antenna is installed on the centerline on top of the fuselage. The aircraft tail structure, wings, engines, and the
fuselage itself could all potentially obstruct the reception or reflect signals arriving from low and negative elevation angles,
which would lead to a loss of lock of the signal, or create local multi-path errors. In order to investigate the possible obstruction
of the signal, we analyzed the number of profiles and their penetration depth as a function of direction relative to the aircraft
heading for the dataset from AR Recon 2021. The predicted maximum number of occultations varies with orientation, therefore
the number of successfully retrieved profiles at different orientations is not comparable. We use the proportion, $P_{alt}$, of the
predicted occultations in the azimuthal bin that have the lowest actual tangent point in the stated altitude range as a proxy for
performance. Summing $P_{alt}$ over all altitude ranges gives the recovery ratio at that azimuth. We try to identify any possible
anisotropy in this proportion.

The heading of the aircraft was simply deduced from the position changes without considering the influence of crosswinds,
which possibly introduces an error in the true heading of a few degrees. The occultation events typically last about 10–15 min,



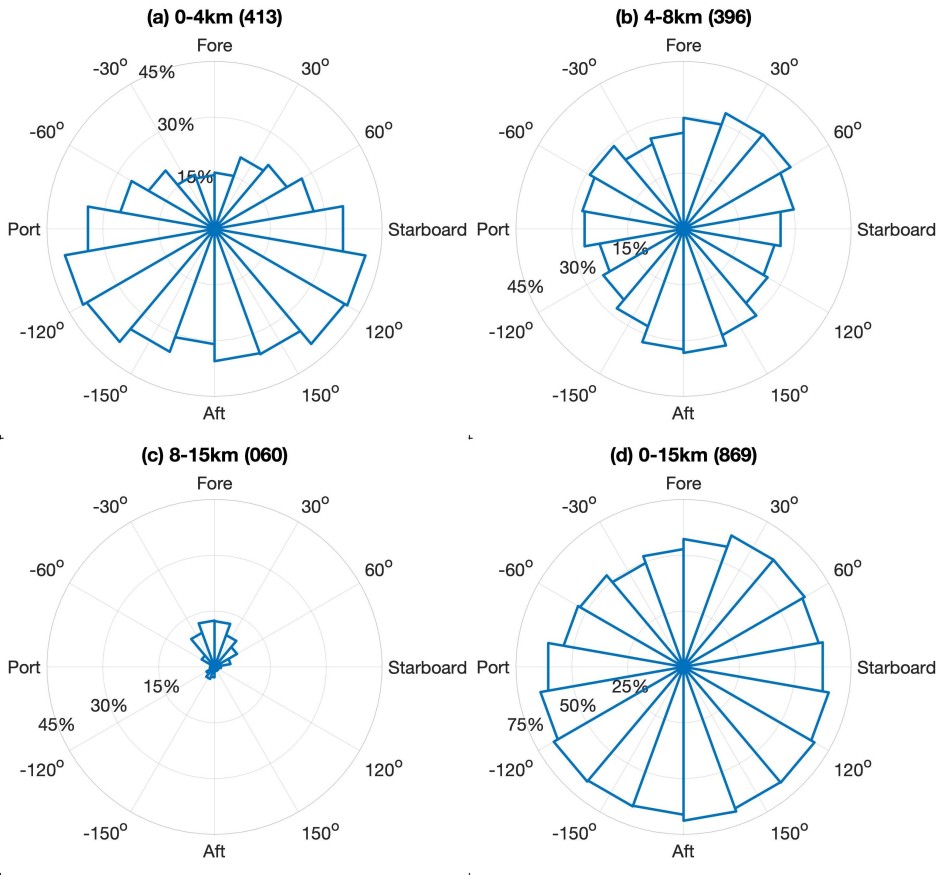

**Figure 12.** Recovery ratio (see text for definition) at different directions relative to the aircraft heading for all ARO profiles in 2021, grouped by lowest tangent point altitude within the range of (a) 0–4km, (b) 4–8 km, (c) 8–15 km and (d) 0–15 km. The range of 0–15 km covers all the retrieved profiles. The maximum radius representing the recovery ratio in polar histograms is 75% for (d) and 45% for (a)–(c). The numbers in the parentheses are the numbers of ARO profiles in this category.

during which time the angle between the aircraft heading and the ray path orientation might change slightly but is negligible relative to the size of the azimuthal bins. For each occultation, we calculated the azimuth of the ray path at the lowest altitude where any potential interference with the signal would be most likely, and calculated the difference with the aircraft heading.

Figure 12 shows $P_{\text{alt}}$ for different directions relative to the aircraft heading, with 0° at fore, ±180° at aft, 90° and -90° at starboard and port directions. The occultations are grouped by their lowest tangent point altitude. The deepest profiles whose lowest tangent points are below 4 km were found to be more likely to be recovered from the aft of the aircraft. About 40% of the predicted occultations in the azimuthal bins in the left and right rear quadrants had tangent points found below 4 km. In contrast, there is a much lower proportion $P_{\text{alt}} \sim 15\%$ for 0–4 km in the fore direction. The profiles with lowest points above 8 km are mainly from the fore direction and their $P_{\text{alt}}$ is less than 10%. Profiles with the lowest tangent points between 4–8 km show a more isotropic distribution, with a slightly higher $P_{\text{alt}}$ in the front right quadrant. If all altitude ranges are considered





(Fig. 12(d)), $P_{\mathrm{alt}}$ sums to a recovery ratio of about 60% at most azimuths, with a slightly lower recovery ratio of about 50%
in the left front quadrant. Considering the installation location of the antenna, the longitudinally extensive fuselage and tail

could potentially block low-elevation signals. The aircraft flew with a constant $\sim 4°$ pitch angle at cruise altitude. Therefore,
the GNSS antenna is tilted backward, making the blockage by the front part of the fuselage more severe. This is the main
reason that the shallow profiles are retrieved from the fore and deep ones are retrieved from the aft and sides. Similar azimuthal
distributions are found in the dataset from AR Recon 2020 when the previous GPS-only antenna was still in use.

Analyzing the possible obstruction could provide some insights into flight planning. Ideally, we should avoid the aircraft

heading toward the most preferential directions of the setting and rising GNSS satellites, such as NW and NE, to achieve the
maximum depth and number of profiles. However, this has not been explicitly considered in the actual flight planning. When
balanced against the flight objectives illustrated in section 3.4 to provide optimally oriented profiles when transecting a SW-NE
trending AR, it would favor flights towards the SE rather than towards the NW, in the anti-velocity direction.

## 4  Quality assessment of the ARO dataset

Haase et al. (2021) estimated the accuracy and precision of the ARO data by comparing multiple co-located ARO profiles from
GPS and Galileo from two different receivers with nearby dropsondes and model reanalysis based on one of the first IOPs
in 2018. The results show an instrumental observation error in refractivity of 0.6% based on the intercomparison of receivers.
The comparison with dropsondes showed excellent agreement with a difference mean of -0.1% and standard deviation of 1.8%.
However, this preliminary statistical comparison was based on 25 profiles from only one flight, including only eight occultations

from Galileo and none from GLONASS. Since 2020, more than 600 ARO profiles have been retrieved each year, together with
more than 400–500 dropsondes launched solely from the G-IV. We have an extensive dataset for a more comprehensive and
robust statistical analysis. In this section, we present results for the ARO profiles with co-located dropsondes and matching
model profiles.

### 4.1  ARO vs. dropsonde

During AR Recon flights, dropsondes were released regularly only along the flight segments that traverse the target area, and
no dropsondes were released during the first and last segments of the flights. ARO profiles were retrieved along the entire
flight track, however they are irregularly distributed and slant away from the track. To compare the refractivity between the
two different datasets, the refractivity was calculated from dropsonde measurements based on Eq. 1 and using the following
conventions

$$p_w = p\frac{r}{\epsilon + r}, \tag{4}$$

where $r$ is the water vapor mixing ratio, calculated from specific humidity $q$ [in kg/kg] through $r = q/(1-q)$, and where
$\epsilon = (R_d/R_w)$ is the ratio of the gas constants for dry air and water vapor, set to 0.622. Although winds blow the dropsonde in
the downwind direction, the drift distance of the dropsonde is much smaller than ARO profiles and was thus neglected when



co-locating ARO-dropsonde pairs. For each dropsonde, we identified all ARO profiles that were within ±20 minutes and 200 km distance. The topmost tangent points of the slant ARO profiles near the flight tracks were closest to the dropsonde profiles,

therefore the location and timing of the highest points of ARO profiles were used to identify the pairs. The possibility exists for multiple ARO profiles to match a given dropsonde, thus they are counted as different pairs. As shown by the map in Fig. 6, an ARO profile that is close to one dropsonde at the top might drift toward a different dropsonde at a lower altitude. We did not specifically consider this condition in the statistical comparison for simplicity, so the resulting error estimate is an upper bound. There are 482 valid dropsondes from the G-IV flights in AR Recon 2021, for which 581 co-located ARO-dropsonde

pairs are identified. Most are within ±10 min of each other (Fig. 13(b)), which is close to the dropsonde release interval. For a matching pair, the distance between the ARO tangent point and dropsonde measurement point is always closer at the top than at the lowest tangent point. The distance at the top is usually within 50 km (Fig. 13(d)). The distance from the dropsonde to the mean location of all the ARO profile tangent points is typically about 100-300 km (Fig. 13(c)). The mean refractivity difference between the two datasets is less than 0.5% and the SD decreases from 3% at 4 km to 1.2% at 8 km (Fig. 13(a), 4). Below 4

km, the SD increases to 7%. Some artifacts exist in the top 500 m due to the binning of profiles with different maximum altitudes, which varied by ±1 km over all the flights, into evenly-spaced grids. The ∼ 400 km horizontal drift distance of ARO profiles is half the typical AR width of 800–1000 km. The two types of measurements may sample areas separated by very strong horizontal gradients, approaching 25%, especially near the edges of the AR. Haase et al. (2021) showed such a case where nearby ARO and dropsondes profiles varied depending on the horizontal gradient of the temperature field, so

there is likely a large contribution to the standard deviation due to horizontal variability of the atmosphere. Considering the slant character of the ARO profiles, the 1.2–7% SD overall indicates the ARO data achieve a very good agreement with the co-located dropsondes. For reference, COSMIC-2 SRO profiles compared to in situ soundings have an SD of 1% to 5.5% over the height range from 2–8 km for a dataset that is not preferentially sampling highly variable storm environments (Ho et al., 2020).

## 4.2 ARO vs. ERA-5


The European Centre for Medium-Range Weather Forecasts (ECMWF) Renalysis 5 (ERA5; Hersbach et al. (2020)) incorporates vast quantities of observations (including the dropsondes from all AR Recon flights) into global estimates of the atmospheric state using advanced modeling and data assimilation systems. The hourly ERA5 reanalysis product on pressure levels was chosen for the comparison, mainly for the high spatial and temporal resolution. It has 37 pressure levels in the vertical, up

to a top level of 1 hPa with a resolution of 25 hPa, equivalent to ∼ 500 m, in the lower troposphere. It has a horizontal resolution of $0.25° \times 0.25°$ (∼ 25–30 km). To find a matching ERA5 profile for each ARO profile, the horizontal drift is taken into consideration. First, the geopotential and geometric height are calculated for each pressure level using the method employed at the ECMWF (Simmons and Burridge, 1981; Trenberth et al., 1993). At each height, the model grid point that is closest to an individual tangent point of the ARO profile was found. Then, the pressure was interpolated logarithmically between the two

nearest levels to the height of the ARO tangent point. The temperature and specific humidity were interpolated linearly in the vertical to the ARO tangent point height. The refractivity was calculated based on equation 1. The final result was an ERA5





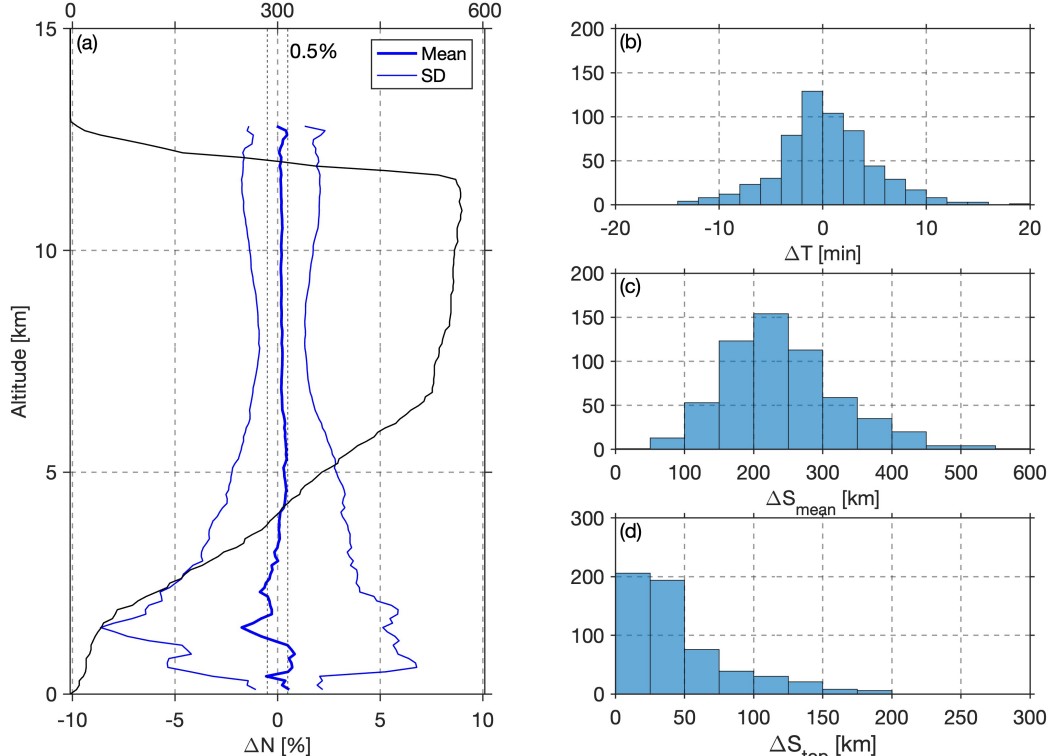

**Figure 13.** (a) Mean and standard derivation (SD) of the percentage difference between the colocated ARO and dropsonde refractivity (ARO minus dropsonde) in 2021. The top axis and black curve denote the number of colocated ARO-dropsonde pairs at each altitude. Vertical dashed lines near zero indicate 0.5% for reference. Histograms of (b) temporal difference, (c) spatial separation between the dropsonde and the mean tangent point position, (d) spatial separation between the dropsonde and the highest tangent point position.

profile that drifts horizontally and contains the same number of points as the given ARO profile. As shown in Fig. 14 and Table 4, the mean difference in refractivity is less than 0.5% and the SD is less than 1.5% above 4 km, and the minimum SD is 1% at 8 km. Below 4 km, the SD increases to 2.8%. There is no clear difference among the occultations of different constellations.

In both comparisons shown in Figs. 13, 14, and Table 4, the ARO refractivity shows a slight positive bias relative to both dropsondes and ERA5, above 5 km. This similarity is likely because the ERA5 already assimilated the dropsonde measurements. There is a negative bias below 4 km that reaches 2%, similar to that seen in SRO, commonly attributed to super-refraction in the lowest troposphere. The ARO observation errors in Table 4 are provided as a guide for data assimilation experiments using ARO refractivity.



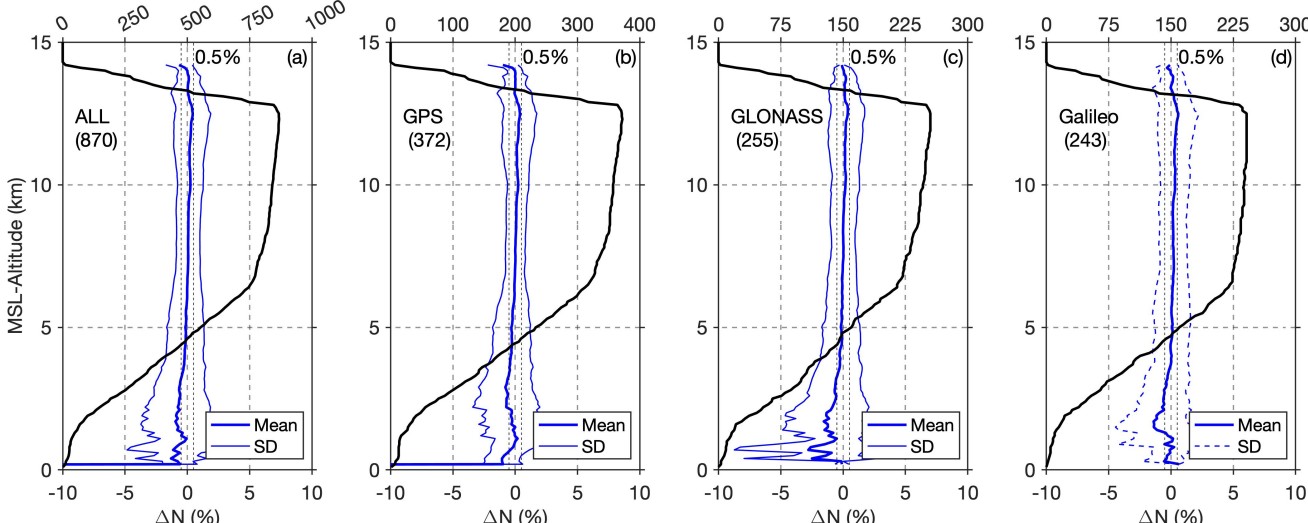

**Figure 14.** Mean and standard derivation (SD) of the percentage differences between the ARO and matching ERA5 reanalysis refractivities (ARO minus ERA5) for occultations retrieved from (a) all, (b) GPS, (c) GLONASS, and (d) Galileo satellites in 2021. The top axes and black curve denote the number of data points at each altitude. The numbers in the parentheses are the total number of profiles and the vertical dashed lines around zero mark the 0.5% for reference.

## 5   Discussion

The previous sections describe the unique characteristics of ARO observations. The height range, temporal-spatial sampling, and general advantages and disadvantages are discussed to guide their use, especially for data assimilation in numerical weather prediction models. Some of the considerations for their current and future use are described below.

The current version of the ARO dataset provides the highest accuracy between 3 km and flight level at ∼ 14 km. Observations in this height range are necessary to reduce initial condition errors that contribute to AR forecast uncertainty (Zheng et al., 2021). For example, Baumgart et al. (2018, 2019) quantified upscale error growth from initial condition errors from convection and latent heating, upper level divergence from these moist processes into the tropopause region, and subsequent organization at Rossby wave scales that contribute to upper-level, near-tropopause PV anomalies. PV anomalies also influence the evolution of extratropical cyclones and their interactions with associated ARs (Zhang et al., 2019). Thus, observations in this critical height range, especially over a broader area than possible with dropsondes alone, can contribute to understanding the interactions of ARs with large-scale dynamics and improving forecasts (Zheng et al., 2021).

This motivates deploying the ARO system to provide extra data for assimilation into numerical models to improve AR forecasts. SRO data assimilation in the ECMWF model shows global positive impact in short term forecast verification against sounding data. The improvement has been predominantly in the stratosphere, and as the density of SRO observations increases, their assimilation has begun to show a small positive impact in 12 hour forecasts in the range 200–300 hPa (12–9 km), however



**Table 4.** Difference (mean and SD) between ARO and dropsondes, and between ARO and ERA5 reanalysis. NaNs in the table mean no data or not enough data samples at that altitude. For reference, the difference between SRO and ERA5 is taken from Murphy and Haase (2022, Table S1), containing current operational space-borne RO missions excluding COSMIC-2.

| Height (km) | ARO vs. dropsonde | | ARO vs. ERA5 | | SRO vs. ERA5 | |
| --- | --- | --- | --- | --- | --- | --- |
| | Mean (%) | SD (%) | Mean (%) | SD (%) | Mean (%) | SD (%) |
| 19.50 | nan | nan | nan | nan | -0.0815 | 0.6845 |
| 18.50 | nan | nan | nan | nan | 0.0410 | 0.4900 |
| 17.50 | nan | nan | nan | nan | 0.3216 | 0.8210 |
| 16.50 | nan | nan | nan | nan | 0.0248 | 0.4532 |
| 15.50 | nan | nan | nan | nan | 0.0157 | 0.4804 |
| 14.50 | nan | nan | nan | nan | -0.0304 | 0.4458 |
| 13.50 | nan | nan | 0.0479 | 1.0075 | -0.0036 | 0.4472 |
| 12.50 | 0.2679 | 1.4860 | 0.3351 | 1.4444 | 0.0652 | 0.4471 |
| 11.50 | 0.1610 | 1.8899 | 0.2251 | 1.3174 | 0.0923 | 0.4266 |
| 10.50 | 0.2351 | 1.7333 | 0.2242 | 1.0773 | 0.0668 | 0.4071 |
| 9.50 | 0.1644 | 1.4140 | 0.1320 | 0.9994 | 0.0798 | 0.4313 |
| 8.50 | 0.1960 | 1.1498 | 0.0833 | 0.9433 | 0.0397 | 0.4460 |
| 7.50 | 0.2313 | 1.1858 | 0.0748 | 0.9496 | -0.0042 | 0.5496 |
| 6.50 | 0.2272 | 1.5472 | 0.0000 | 1.1123 | -0.0485 | 0.7219 |
| 5.50 | 0.4292 | 2.2042 | -0.1141 | 1.3568 | -0.0876 | 0.9143 |
| 4.50 | 0.3899 | 2.9033 | -0.1366 | 1.4913 | -0.1092 | 1.1851 |
| 3.50 | 0.1047 | 3.4255 | -0.3246 | 1.7352 | -0.1338 | 1.4475 |
| 2.50 | -0.5716 | 4.4028 | -0.6373 | 2.2687 | -0.1968 | 1.8909 |
| 1.50 | -1.7480 | 6.8806 | -0.7164 | 2.8196 | -0.6087 | 2.6573 |
| 0.50 | nan | nan | -0.9389 | 1.7439 | -1.7611 | 2.7521 |

still less than 1% improvement in the range 300–750 hPa (3–9 km) (Ruston et al., 2022). Given the favorable prospects from preliminary examples of ARO impact assessment (Chen et al., 2018), it is likely to be beneficial to assimilate ARO data that densely sample high impact weather in the mid-troposphere in routine operations, in near real-time. The recent development of assimilation operators that are tailored for ARO data is an important achievement that enables this (Hordyniec et al., 2024), as well as the increase in the number of ARO observations with the expansion of AR Recon (Lavers et al., 2024).

ARO provides direct measurements of refractive bending angle and derived refractivity. Both variables depend on the combination of pressure, temperature, and moisture, which cannot be uniquely determined. In the upper troposphere above 9 km, the effect of moisture is negligible, and using the hydrostatic equation, the pressure and temperature can be estimated with good accuracy (Kursinski et al., 1997; Cao et al., 2022). However, in the moist lower troposphere, that is not the case. The 1D-var method has been used for SRO observations to derive moisture with prior information from a numerical weather model.





The retrievals and their errors are then dependent on the first guess model (Poli et al., 2002). In the release of the ARO dataset presented in this study, we limit the products to bending angle and refractivity, and dry pressure and temperature, to avoid introducing additional error or ambiguity into the products. Therefore, the preferred approach is to assimilate refractivity directly, with a local or non-local operator (Chen et al., 2018), or bending angle using a 2D operator (Hordyniec et al., 2024). Both of these techniques take into account variations of atmospheric structure along the long horizontal ray path when assimilating such that the observations can be used in high resolution models. This property can also help spread out the information from dropsondes and make the analysis less susceptible to small-scale variations that are present in the dropsonde data that are not resolvable by the finite grid spacing of the model.

The ARO profiles presented in this study are retrieved from conventional GNSS receivers utilizing PLL tracking. This type of receiver has the advantage of easy operation and avoids an additional lengthy data pre-processing step. However, they cannot always continuously track the signals penetrating to the lowest part of the troposphere (0–3 km), where atmospheric moisture concentrates. In this altitude range, the GNSS signals can undergo multi-path propagation due to sharp gradients in moisture, leading to the measured signal not representing a single ray path but a combination of multiple rays arriving at the receiver simultaneously (Sokolovskiy, 2003). Only about 20–30% of the profiles in the current dataset have the lowest point below 3 km. The fluctuations of observed excess Doppler at lower altitudes lead to downgraded data quality and sometimes must be discarded. There would be great benefits to extend the sampling to lower altitudes, given that the highest moisture flux in the ARs is concentrated at about 1.5 km altitude (Ralph et al., 2005). The ARO capability using open-loop (OL) tracking is currently under development for the G-IV to reach the same penetration depth as was achieved with earlier prototypes (Haase et al., 2014; Wang et al., 2017). This is expected to provide a dataset with more extensive sampling in the lowest troposphere. Another potential benefit of OL tracking would be improving the recovery of rising occultations. On average, the number of rising occultations is at least 10% less than the setting ones. The lowest tangent point from rising occultations is generally 1–1.5 km higher than the setting ones, regardless of the constellation, antenna, and receiver type. This is not unexpected because the acquisition and tracking of GNSS signals from satellites not in sight (occulted by the Earth) are difficult for PLL receivers. Using OL tracking on pre-recorded raw RF signals with the time-reversed achieves equivalent performance for both setting and rising occultations (Wang et al., 2017).

The most distinct characteristic of ARO is the oblique nature of the profiles, which has advantages and disadvantages. Compared to nearly vertical dropsonde profiles, the horizontal drift introduces complexity in interpreting the information in 3-D space. Neglecting the drift and the extra horizontal interpolation and/or approach to binning potentially results in misleading artifacts. In studies where the slant profiles were compiled to resolve large-scale gravity waves, the obliqueness exerts little influence, and the slant profiles were treated as vertical (Cao et al., 2022). However, DA experiments revealed that the forecasts are sensitive to the positions of profiles (Chen et al., 2018), thus tangent point drift should always be considered. On the positive side, the large drift expands the sensing area further away from the flight tracks such that the flights cover the extended area of high sensitivity. As discussed in section 3.2, the spatial resolution is about 150–250 km along the ray path and on the order of 200–400 m perpendicular to the ray path. The atypical high vertical resolution is an advantage of ARO observations such that it can resolve fine-scale structures. However, the derived refractivity should be treated as a weighted mean over the central





part of the ray path rather than as a point value. In the aforementioned DA experiments, at least two types of operators were used, the standard local refractivity operator and the non-local excess phase operator that allows adjustments to the model at all points along the ray path. The former has the advantage of low computation cost but leads to significant errors near regions with strong horizontal gradients (Chen et al., 2018; Xie et al., 2008).

In current AR Recon operations, the flight planning primarily focuses on obtaining transects of dropsonde traversing the AR over high sensitivity areas within a targeted 6-hr DA window. The ARO observation locations are not explicitly considered in the flight planning. The ARO profiles simply occur in the vicinity of the aircraft flight track in a quasi-random manner. They probe the same high-sensitivity areas as dropsondes and, in addition, cover a large geographic region en route to the targeted areas where no dropsondes are launched. This highlights ARO's advantage as a non-invasive technique that can be used to

obtain valid observations over land and in areas with high air traffic where dropsondes cannot be launched, or where the target area is too dangerous for aircraft to fly over directly.

ARO was first brought to the field to study tropical cyclones in the PREDICT campaign. Although it was limited to a proof-of-concept deployment, the retrieved ARO data showed some positive impacts on the forecast of hurricane Karl (Chen et al., 2018). The NOAA G-IV aircraft routinely executes surveillance and research flights over the Atlantic, Gulf of Mexico, and the

Caribbean during hurricane seasons. The successful deployment of ARO in AR Recon missions can be expanded to provide critical information for hurricane forecasts and research. We have deployed the ARO equipment on the G-IV aircraft during the hurricane field program (HFP) in the 2020, 2022, and 2023 Atlantic hurricane seasons, and the dataset is available for future hurricane studies.

Considering that most modern aircraft already have one or more GPS/GNSS receivers installed onboard for navigation

purposes, by making some minor modifications, the receivers onboard commercial aircraft could be utilized to provide a vast amount of ARO data. It would dramatically augment the existing aircraft dataset to expand from in situ flight level measurements to full profiles along the flight. This could potentially improve global weather forecasts by incorporating ARO datasets from commercial aircraft that are flying globally daily, especially trans-oceanic flights over data-sparse oceans (Lesne et al., 2002).

**6   Summary and conclusions**

Advances in modern GNSS technology have brought Airborne Radio Occultation (ARO) from the first experimental GISMOS prototype to the current operational version that regularly flies onboard NOAA and USAF aircraft. The system was deployed on the NOAA G-IV jet during Atmospheric Rivers Reconnaissance missions (AR Recon) in 2018, 2020 and 2021, with an additional piggyback mission in 2019. The final dataset is comprised of $\sim$ 1700 ARO profiles from 39 flights ($\sim$ 260 flight

hours) from multiple GNSS constellations, including GPS, GLONASS, and Galileo. Typically, 30–45 refractivity profiles were retrieved over each 7–8 hour flight, from aircraft cruise altitude (13–14 km) down into the lower troposphere. More than 50% of the profiles extend below 4 km altitude. ARO provides slanted profiles with a vertical resolution better than 400 m that



extends roughly 400 km to the sideway of the flight track, essentially linking dropsonde observations beneath the flight track to mid-level features of the larger scale environment.

To verify the accuracy of the ARO observations, the retrieved profiles were compared to refractivity calculated from the dropsonde data from the same flights and the ERA-5 reanalysis. Good agreement was found with both datasets. The ERA-5 refractivity profiles were interpolated from the original evenly spaced model grid to the drifting ARO tangent point locations. The mean and standard deviation of the difference of the ARO refractivity from ERA-5 were less than 0.5% and 1.5%, respectively, from 4 km to flight level ($\sim 14$ km). The same quality was achieved for occultations from the three different constellations. For

the dropsonde comparisons, the dropsonde was selected to be the closest within 10 min and 100 km of the ARO profile. The mean difference of the ARO refractivity from the closest dropsonde did not exceed 0.5% above 3 km. The standard deviation was less than 1.5% from 6.5 km to flight level ($\sim 14$ km). Below 6.5 km, the standard deviation increased from 1.5% to 4.5% at 2.5 km, primarily because the tangent points were sampling significantly different spatial locations than the dropsondes at lower levels. Given that the horizontal variability of refractivity within the AR structures exceeds 25% (Haase et al., 2021), this

level of agreement confirms their consistency.

AR Recon campaigns are designed to address the observational needs over the data-sparse and cloud-covered oceanic areas associated with ARs to improve understanding of their physics and dynamics. These campaigns are also important for AR forecasting because they provide data to initialize and validate NWP models. The highly maneuverable aircraft was deployed to take direct measurements in the AR environment in areas identified as sensitive regions to forecast errors (Reynolds et al.,

2019). Specifically, flights sample areas with the highest sensitivity where initial condition errors are likely to trigger forecast errors in the landfall location of ARs and the consequent precipitation over the western US. The dropsondes and ARO provide complementary sampling over these target regions by sampling directly beneath and around the flight track, respectively, efficiently using limited flight resources. The high vertical resolution aircraft measurements provide the advantage of snapshot-style observations that fill in the gaps in satellite radiances due to clouds and precipitation and assure sampling in the desired

window regardless of the time sampling of satellite overpasses. The retrieved ARO refractivity anomaly (difference from climatology) captures the important features of ARs (Fig. 11), and as indicated in previous work (Haase et al., 2021, Fig. 8), in particular the low-level high moisture core of the AR. Together with the dropsondes, the AR Recon datasets are available to construct a comprehensive 3-D picture of winds, temperature, and moisture in the target areas.

*Code and data availability.* The four years ARO dataset presented in this study can be downloaded from the UCSD library research data

curation service (https://doi.org/10.6075/TOBEDEFINED) and the Haase group webpage (https://agsweb.ucsd.edu/gnss-aro/ (Haase and Cao, 2024)). The dropsonde data was provided by the Center for Western Weather and Water Extremes and more information about AR Recon campaign can be found on their webpage (https://cw3e.ucsd.edu/arrecon_overview/ (Center for Western Weather and Water Extremes, 2024)). The flight level meteorological data, raw dropsonde files, and flight report of NOAA G-IV were downloaded from the NOAA Office of Marine Operations and Aviation Operations (OMAO) data server (https://seb.omao.noaa.gov/pub/acdata/ (Office of Marine and

Aviation Operations, 2024)). The ECMWF reanalysis ERA5 product was provided by UCAR through the Research Data Archive (https://rda.ucar.edu/datasets/ds633.0/ (European Centre for Medium-Range Weather Forecasts, 2019)). The multi-GNSS satellite orbit and clock


products used for precise positioning and excess phase calculation were downloaded from the GNSS Center at Wuhan University (WHU) (ftp://igs.gnsswhu.cn/pub/whu/phasebias/ (PRIDE Lab/Wuhan University, 2022b)) and the Center for Orbit Determination in Europe (CODE) operated at the Astronomical Institute of the University of Bern (http://ftp.aiub.unibe.ch/CODE/ (Dach et al., 2023)). The software for

precise point positioning is provided by PRIDE Lab at Wuhan University (https://github.com/PrideLab/PRIDE-PPPAR (PRIDE Lab/Wuhan University, 2022a)).

## Appendix A:  ARO data processing procedures

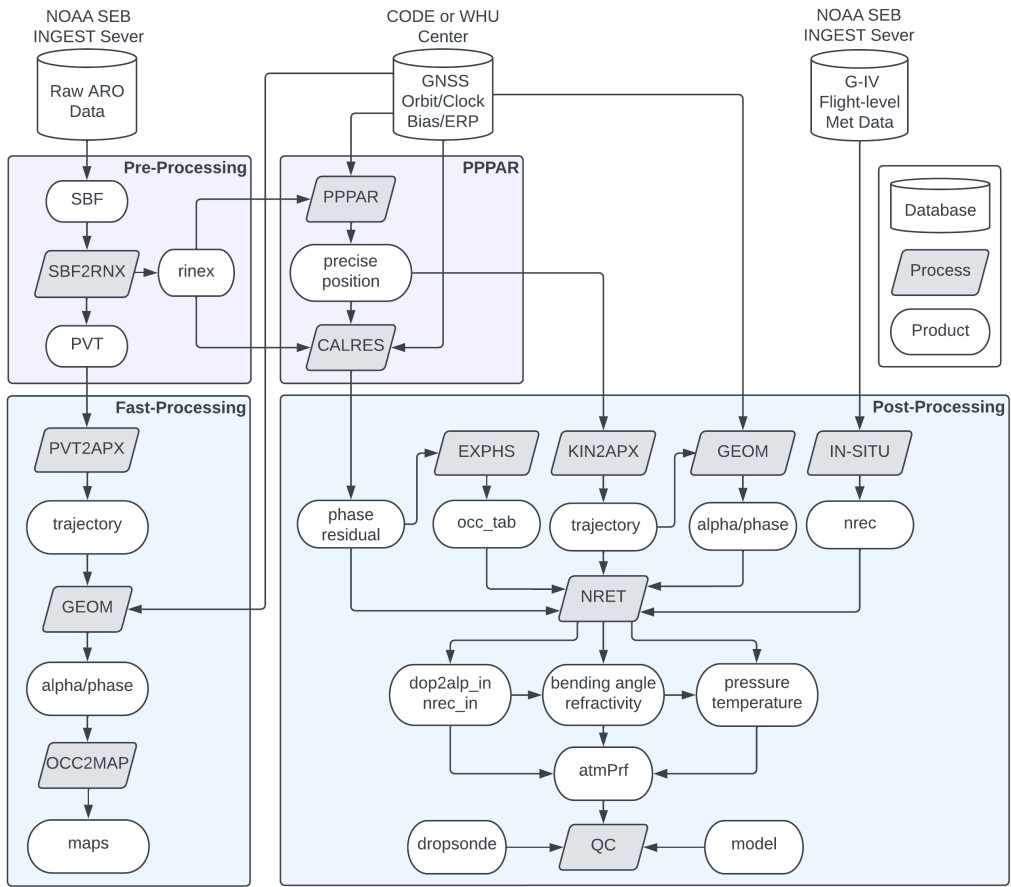

**Figure A1.** A flowchart describing all the steps of ARO data processing, from raw data recovery to final product quality control. The information reflects the most recent version as of 2024. The only difference with what is presented in this study is that raw ARO data in 2018-2021 was downloaded from the equipment by the flight crew rather than downloaded from the NOAA ingest server.

The results presented in this study (2018–2021) are from the experimental deployment of ARO when the raw data were downloaded manually by the aircraft crew members after each flight and/or during aircraft downtime. Beginning in AR Recon





2022, the raw ARO data has been migrated into the aircraft real-time datastream and transferred to the NOAA/OMAO ingest server via SATCOM. Therefore, the raw ARO data has been retrieved from the NOAA data server since then. The full ARO data processing procedures are shown in Fig. A1. The procedures can be divided into four major steps: (I) Pre-processing: download, clean, concatenate and reformat raw ARO data. (II) PPPAR: calculate the precise positions of the aircraft and phase residuals with GNSS satellite orbit/clock/bias/quaternion products. (III) Fast-processing: simulate all possible occultation locations given aircraft trajectory and GNSS satellite ephemerides. (IV) Post-processing: retrieve the final bending and refractivity profiles and generate standard ARO products. In typical ARO workflow, the "pre-processing" and "fast-processing" are generally executed either in near real-time or within a few hours of the end of the flight. They provide an overview of raw data quality and a snapshot of the ARO product spatial coverage. The "PPPAR" and "post-processing" are executed either after 24 hours when rapid GNSS orbit/clock products become available at the WHU GNSS analysis center or after two weeks when final GNSS orbit/clock products become available at the CODE GNSS analysis center.

At different stages in the processing, several QC procedures are applied to improve the accuracy and the recovery of more profiles. We first verify that flight-level meteorological measurements are complete and there are no outliers, before the ARO retrieval is initiated. There may exist a small trend in the excess Doppler, likely due to unmodelled errors during the position and phase residual calculations. When approaching the zero elevation angle above the horizon, the bending angle derived from the segment of data at positive elevation must match the bending angle for the negative elevation. If any mismatch is detected, then the excess Doppler is adjusted for the individual profile. This greatly reduces the error at the top of the profile, where the accumulated delay is relatively small. The definition of each process is provided below.

- **SBF2RNX**: convert raw Septentrio Binary Format (SBF) files into rinex and ASCII format.

- **PVT2APX**: reformat receiver PVT (position, velocity, and timing) solution to standard trajectory file.

- **GEOM**: get the geometry of occultations based on aircraft trajectory and GNSS satellite ephemerides, and simulate the bending angle and excess phase in a given climatological atmosphere as defined by the CIRA-Q model.

- **OCC2MAP**: create maps with predicted occultations.

- **PPPAR**: Precise Point Positioning with Ambiguity Resolution to determine aircraft positions.

- **CALRES**: calculate the phase residuals with the aircraft positions fixed.

- **KIN2APX**: reformat PPPAR kinematic solution to standard trajectory file.

- **EXPHS**: sort the phase residuals and find the pair of GNSS satellites, one at the low elevation occultation position and one at high elevation, to apply single-differencing to remove receiver clock errors.

- **NRET**: calculate the bending angle and retrieve the refractivity profiles from the Abel transform.

- **IN-SITU**: calculate the in situ refractivity from the aircraft flight-level meteorological measurements.





– **QC**: quality control of the final products.

Detailed descriptions of each type of data files are provided below. The data formats are either universal, equipment manufacturer-defined, or model provider-defined. Some intermediate data formats are self-defined.

- **SBF**: Septentrio Binary File, a binary file in the manufacturer-defined format generated by Septentrio receivers.

- **PVT**: Position, Velocity, and Timing solutions estimated by the Sepetentrio receiver in real-time but less accurate.

- **rinex**: a standard format for GNSS observables, in either version 2 or version 3.

- **trajectory**: a time series of positions, velocities, attitudes of the aircraft, and uncertainties of all variables.

- **alpha/phase**: a time series of simulated bending angle and excess phase.

- **precise position**: a time series of accurate aircraft positions determined by PPPAR.

- **phase residual**: a time series of calculated phase residual with aircraft position fixed, containing receiver clock errors.

- **occ_tab**: a table of the pairs of satellites for single-differencing, one at zero elevation and one at high elevation.

- **nrec**: a time series of aircraft flight-level (in situ) meteorological measurements, inclduing refractivity.

- **dop2alp_in**: a time series of the aircraft and satellite positions and velocities, and excess Doppler for each occultation.

- **nrec_in**: a time series of in situ refractivity for each occultation.

- **bending angle/refractivity**: profiles of calculated bending angle vs. impact parameter and refractivity vs. altitude.

- **pressure/temperature**: profiles of derived atmospheric pressure and temperature, based on dry atmosphere assumption.

- **atmPrf**: final product in the format defined by the COSMIC Data Analysis and Archive Center (CDAAC).

- **dropsonde**: dropsonde observations.

- **model**: forecast or analysis products from ECMWF/ERA5 or NCEP Global Forecast System (GFS).

- **maps**: a map showing aircraft trajectory and all possible occultations, with modeled IVT illustrating ARs.

*Author contributions.* BC deployed the instruments and collected, processed, and analyzed the data. BC formulated and wrote the manuscript, and prepared the figures, with contributions from all co-authors. JH conceived and acquired funding for the campaigns as principal investigator. JH assisted in equipment deployment and data collection and guided the research investigation. MM executed the simulations and provided all the model-related products. AW managed and coordinated the overall operation of the AR Recon campaign and collection of dropsonde data.



*Competing interests.* The authors declare that they have no conflict of interest.

*Acknowledgements.* This work was supported by NSF Grants AGS-1642650 and AGS-1454125, and NASA Grant NNX15AU19G. Support was also provided through the UCSD Center For Western Weather and Water Extremes (CW3E) Atmospheric River Research Program from the California Department of Water Resources and from the US Army Corps of Engineers. The authors sincerely acknowledge the continued support from NOAA Aircraft Operation Center, in particular, G. Defeo, J. Parrish, A. Lundry, and L. Miller for assisting with installation and
operation of the ARO equipment on the G-IV aircraft and implementing the SATCOM transfer of the ARO data. The authors thank CW3E, NCEP, and all of the collaborators in the AR Recon Research and Operations Partnership for providing the opportunity for ARO observations during AR Recon flights. The authors would like to thank the NSF/NCAR EOL facility for the loan of the Applanix GPS/INS system, and A. Borsa of SIO/UCSD for the loan of the Septentrio PolaRx5 receiver in 2018. J. Johnson and J. Dahlberg of NOAA National Geodetic Survey are acknowledged for their support the piggyback deployment of the ROC2 receiver in their 2019 Grav-D survey flights and for
operating the PwrPak-7 GNSS/INS system in 2021. The development of the ROC2 receiver was supported by NSF grant AGS-1642650 for the Strateole-2 project, with help from D. Jabson, J. Souders, and S. McPeak of SIO/UCSD. M. J. Alexander and M. Bramberger of NWRA are acknowledged for the meaningful discussions about Strateole-2/ROC2 data processing, which was later migrated into the standard ARO data processing.



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
