# Peer review of "Observing atmospheric rivers using multi-GNSS airborne radio occultation: system description and data evaluation"

_Atmospheric Measurement Techniques, 2024_

## Referee Comment (RC3)

First Round of Review of Manuscript AMT-2024-119 Submitted to Atmospheric Measurement Technology

Manuscript Title: Observing Atmospheric Rivers using Multi-GNSS Airborne Radio Occultation: System Description and Data Evaluation

Corresponding Author: Jennifer Haase (jhaase@ucsd.edu)

**Study Summary:**

This manuscript describes the use of airborne GNSS radio occultation for observing atmospheric river (AR) events that impact the west coast of the United States. These airborne radio occultation (ARO) observations are shown to be successful in observing AR events due to the inherent ability of ARO profiles to ignore clouds and precipitation, resulting in data that can observe the thermodynamics of ARs where other remote sensing methods fail due to low vertical resolution or signal attenuation. A ARO full observation and retrieval system is described, and retrieved results were compared to ERA5 model reanalysis output as well as in-situ dropsonde observations. Mean refractivity differences between ARO profiles and ERA5/dropsonde profiles was found to be less than 0.5% magnitude above 3 km with varying standard deviation that is higher at lower altitudes, indicating the high quality of the observations and their potential usefulness in numerical weather prediction of AR events.

**General Comments:**

This manuscript is generally well-written and provides very unique data and results. However, my major comment regarding the manuscript is that it is quite long, with more specific provided as part of my comments. Overall, I would recommend publishing this paper after the below comments and suggestions are sufficiently addressed. It is likely that I will not catch all errors, so I would encourage additional read throughs to find any additional issues. My overarching notes for this study are the following:

1. General Readability/Structure and Grammar:
   a. In general, I would avoid the use of phrases like "we implemented" or "our tests" in regard to the experiments that took place. Pronouns are generally not used in technical writing.
2. Introduction and Motivation:
   a. The introduction is generally well-written with only a few issues to fix detailed in the line-by-line comments.
   b. Based on a Google Scholar search, the authors do not mention recently published work on engineering of new airborne RO payloads and the use of commercial aircraft for airborne radio occultation (e.g., Chan et al., 2022; Xie et al., 2024). This would be most relevant to the discussion around line 110. Do the authors have a specific reason they chose not to include this

potentially relevant information? If not, I would encourage them to work relevant references into their introduction

3. Methodology:
   a. I was not able to find the authors' Equation 1 in (Rüeger, 2002) or (Rüeger, 2002). I am concerned that the authors are not using the correct formulation for refractive index and refractivity. The authors' equation 1 is especially concerning because the second term in Eq. 1 actually removes some of the contribution of water vapor rather than the traditionally-documented addition effects of water vapor to atmospheric refractivity. The authors will need to either fix this error or adequately justify the use of this different equation.

4. Results and Discussion:
   a. The first few sections of the results are primarily dedicated to various retrieved profile properties such as resolution, penetration depth, and obliqueness. While I understand the authors' desire to be complete in their analysis, it seems to me that while this information is unique to AR events, this type of information has been described by previous studies that develop the methodology. I would suggest trimming and/or cutting some of the material regarding resolutions, durations, etc. so that the authors can focus more on the observations themselves and their impact on NWP simulations of AR events, as the purpose of the manuscript purports to be focusing on based on the title an abstract.
   b. The analysis of the effect of the azimuth of the RO relative to the aircraft (e.g, Figure 12) is very interesting. Is the rate of obstruction due to the aircraft something that could be resolved with additional antennae? Or perhaps switching to open-loop tracking? It seems like the highest rates of obstruction occur for the most impactful ARO profiles, so future solutions to this would represent a significant increase in the quality of the ARO data.
   c. How do the authors correct for errors in bending near the aircraft (receiver)?
   d. Are the authors down-sampling the ARO profiles to the ERA5 vertical resolution? It is not clear to me, because the authors state they use the pressure-level ERA5 data but then refer to the model grid points in line 593. I am concerned that there would be significant loss of information from the ARO as a result of down-sampling the ARO. Model level data would be more sufficient for this comparison, particularly in the lower troposphere. Additionally, is linear interpolation of the ERA5 temperature and humidity sufficient? Why not use a higher-order interpolation?
   e. I would suggest that the authors do more to put their results in context with other recently published in-atmosphere RO studies, particularly those that occur in high-moisture environments.
   f. While the ARO data from AR recons undoubtedly provide significant information about the atmospheric state, it is not clear to me how much the ARO profiles, which generally do not penetrate below 4 km due to closed-loop tracking, actually penetrate into the clouds and precipitation resulting

from AR. Can the authors comment on what the profiles that penetrate below 4 km show in regard to the regions that are cloudy and/or actively precipitating? I feel like this information would contribute more to the manuscript overall than the long discussion of various resolutions, penetration, etc. of the profiles.

5. Conclusions:
    a. Does the AR dataset have a formal citation? What about a DOI? Archiving the data with NOAA would allow for the creation of both of these things fairly easily. The data should have a proper citation regardless, but archiving with NOAA is merely a suggestion based on reviewer experience.
    b. Again, the comparisons of the ARO profiles with dropsondes and ERA5 are really not properly put into context with more recent studies. I strongly suggest that the authors do a more thorough review of recently published work to compare their statistics with other observations from SRO and ARO, particularly if high-moisture observations are available.

Please see my line-by-line comments for more specific details.

**Line-by-Line Comments:**

1. L013: "Lock" is a little bit general. I assume the authors mean signal phase lock. Please consider referring specifically to signal phase lock.
2. L026-027: Please reword this sentence to remove repeated occurrences of "on the other hand" within the same sentence. I would suggest at least two sentences from this one.
3. L144: I would advocate the use of an Oxford comma here. Specifically, after "transmitter (satellite)…".
4. L241: There should be an Oxford comma after "and" here for lists.
5. L249: "… relativity effect" should be "relativity effects" here.
6. L250: Should "effort" be "effects" here? Also, is this simply the first-order ionospheric correction? Was any testing done using higher-order corrections? If not, what would be the expected effects, if any?
7. L262-263: I would specify that the "positive bending angle" and "negative bending angle" are really "positive elevation angle bending" and "negative elevation angle bending" (or something similar) throughout the paper. Otherwise, it implies that the bending angle itself is negative, which is not physically consistent. This should be true throughout the manuscript.
8. Figure 3; What is the reason for the color shading? The link to the AR Recon data in the caption of Figure 3 should be formatted into a proper citation as a dataset to be consistent with AMT regulations.
9. L332-333: The description of the colored lines in Figure 4 should probably be limited to the figure caption.
10. L350: I don't believe that "decimated" is the correct word here. Perhaps the authors meant "delineated"?

11. L404: I would be careful describing this as "tomographic-style". I'm not entirely convinced that this would fit the traditional tomography definition.
12. L527: I would change the end of this sentence to be " ...starboard and port directions, respectively."
13. L545: "co-located" should be "colocated" to be consistent with journal hyphenation rules and consistency. Typically, "collocated" or "colocated" are the journal-recognized spellings. This should be changed throughout the manuscript.
14. L550-552: Please re-word these sentences to remove the second sentence. Perhaps something like "... solely from the G-IV, resulting in a much more extensive dataset..."
15. L554: "dropsonde" should be plural here
16. L556: "ARO profiles ... from the track" is not needed as it was heavily discussed in previous sections.
17. L560: I think there is an indent here where there shouldn't be one.
18. L596: Please see general comment #2a regarding the calculation of atmospheric refractivity. This is likely creating errors due to the incorrect refractivity formulation.
19. Table 4: I would suggest replacing the "nan" values with something to indicate that there is simply no data there. Maybe a dash?

**Reviewer References:**

Chan, B. C., Goel, A., Kosh, J., Reid, T. G. R., Snyder, C. R., Tarantino, P. M., Soedarmadji, S., Soedarmadji, W., Nelson, K., Xie, F., and Vergalla, M.: Commercial GNSS Radio Occultation on Aerial Platforms With Off-The-Shelf Receivers, navi, 69, navi.544, https://doi.org/10.33012/navi.544, 2022.

Rüeger, J. M.: Refractive Index Formulae for Radio Waves, XXII International Federation of Surveyors, Washington, DC, USA, Integration of Techniques and Corrections to Achieve Accurate Engineering Survey, 13, 2002.

Rüeger, Jean M.: Refractive Indices of Light, Infrared, and Radio Waves in the Atmosphere, University of New South Wales, 2002.

Xie, F., Nelson, K. J., Chan, B. C., Goel, A., Kosh, J., and Vergalla, M.: First Results of Airborne GNSS Radio Occultation Sounding From Airbus Commercial Aircraft, Geophysical Research Letters, 51, https://doi.org/10.1029/2024GL110194, 2024.

---

## Author Response (AR1)

We sincerely appreciate the reviewers' thoughtful comments and suggestions on our manuscript. We have carefully addressed each point, with our responses highlighted in blue and bold in the following sections. A summary of the revisions made to the manuscript is also attached below for some comments and highlight in red and bold. In the revised manuscript, a major change is that several sections of technical information, such as the ARO equipment specifications, are moved to the Appendix, while the overall manuscript structure remains the same.

Reviewers' comments and suggestions: black

Our responses: bold and blue.

Our revisions: bold and red.

**Reviewer #RC1 Sean Healy**

**General comment**

This is a useful paper on airborne radio occultation measurements, and I recommend publication after minor corrections/clarifications outlined in the specific comments below.

**Specific comments**

Page 2, line 36: The data gap identified by Zheng (2021). Does this account for the information provided by COSMIC-2 since 2020, or the fact that many NWP centers can assimilate microwave radiances in an all-sky framework?

**We added the text:**

Even as new techniques for all-sky radiance assimilation are being tested to improve their use in cloudy areas (Li et al., 2022), it has been shown that reconnaissance data can improve the initial state such that more all-sky radiances pass the quality checks (Zheng et al., 2024).

Zheng et al. 2021 did not include data from the "newly launched" COSMIC-2. For reference, the coverage of COSMIC-2 is discussed at line 110:

An average of 3 profiles within the highest resolution domain (11° by 11°) per cycle (6 hours) are assimilated, which yields a modest 10% intensity forecast skill improvement for several lead times.

**We added comparable numbers for ARO at line 117:**

"Numbers are increasing in the mid-latitudes with the launch of recent commercial satellite constellations, however ARO focuses on the localized storm environment, with a dense distribution of observations on the order of 55 profiles within a 6 hour window in a comparable sized domain, so it is more likely to capture a sensitive area that could impact the downstream evolution of a particular storm event."

Page 4, line 105. Why is a 10 % improvement in TC intensity forecast with COSMIC-2 described as "modest"? This leads to a more general point. How does the accuracy of the SRO and ARO observations compare in the troposphere? ARO may have better sampling, but my understanding is the accuracy is poorer than SRO. Is that correct – please quantify/discuss?

The terms "10%" and "modest" are directly quoted from Miller et al. (2023): "It is shown that COSMIC-2 assimilation yields a modest 10% intensity forecast skill improvement for several lead times, although more substantial intensity forecast improvement is found for a few

**forecasts where the COSMIC-2 observation assimilation helps correct a lower-to-mid tropospheric water vapor bias."**

ARO is generally less accurate than SRO in the upper and middle troposphere, primarily due to the turbulent motion of the aircraft. We have included a dedicated section discussing accuracy and comparing ARO with SRO in figures 13 and 14.

Equation 1: I do not think Rueger (2002) recommends 77.6 for the first term. Please clarify.

That was an error in the text. We have confirmed that the first coefficient is 77.689 and have also double-checked that the refractivity calculated from both the model and dropsonde products is based on the published Rueger (2002) coefficients.

Page 6, line 146. In SRO, the Doppler shift to bending angle step is performed assuming the refractive index at the receiver is unity. Is this assumption made for ARO? Please clarify the text. In addition, are you performing a geometrical optics retrieval rather than a wave optics retrieval to bending angle?

We do not apply the unity assumption, as the aircraft operates within the neutral atmosphere. Instead, we use the aircraft in-situ observations to calculate the flight-level refractive index, which is then used in the geometrical optics method to retrieve the bending angle and impact parameter for each ray path. We also use the flight-level refractive index in the Abel transform. We have explicitly mentioned the "geometrical optics assumption" when discussing vertical resolution in a later section. To enhance clarity, we have defined this at the first usage (line 161) and revised the sentence at 164 accordingly.

Line 161: "where \$n\_R\$ is the refractive index at the aircraft location"

Line 164: "The bending angle and impact parameter are derived from the observed Doppler shift and position and velocities of aircraft and satellites based on the geometric optics assumption, using the flight level refractive index (Vorobev1994, Born1999)."

Top of page 7: The point about assimilating refractivity and handling the ambiguity there is correct, but it should be added that most NWP centers assimilate SRO as bending angle profiles.

In this section, we state the possibility of retrieving moisture information from the refractivity in equation (1) using variational methods. We removed the sentence about the assimilation of refractivity into the model, and we dedicate a discussion of the DA efforts regarding RO data in section 5.

As mentioned above, the retrieved refractivity, with its combined effects of the hydrostatic and moisture terms, can be used directly to constrain NWP models through variational data assimilation. Page 7, line 178: "in situ measurements used in the ARO retrieval". As above, just in the Abel transform or in the Doppler to bending angle step as well?

The in situ/flight level refractive indexes were used in both bending angle retrieval to estimate impact parameters for each ray path and Abel inversion to retrieve refractivity. To avoid confusion and repetition, we removed the sentence, but we added designated discussions in the methodology section of how in-situ measurements were used in the retrievals.

Page 10, line 250: Why is the ionospheric correction handled at the phase level? For SRO, this is known to be less accurate than bending angle, but does the use of partial bending angles mitigate this potential error?

For ARO geometry, the bent ray path is very close to the aircraft and has a shorter distance than SRO. The separation of the L1 and L2 ray paths is much less than that of SRO, especially within the troposphere. In the past, we tried the ionospheric correction in the bending angle; the final results showed similar quality, and the improvement was very small in the ARO retrievals.

Page 11, line 293, L1 only: Healy et al (2002) suggest a single frequency is sufficient for partial bending angles. Is that not correct in practice?

It is theoretically possible to use the single-frequency bending angle. When calculating the partial bending angle, for a spherically symmetric ionosphere, the ionospheric error would be eliminated when subtracting the positive from the negative elevation bending angle to retrieve the partial bending angle. However, the raypath may be traveling through a non-spherically symmetric ionosphere with short wavelength variability.

For ARO, we use the same raw data to do the "orbit" (aircraft trajectory) and RO calculations. In practice, the limiting factor is that our current POD processing software (PPP, precise point positioning) requires dual frequency data to solve for the aircraft positions, and also solves for clock offsets of the phase measurements. To facilitate the application of the clock corrections, we use the same ionosphere correction to the excess phase results as were used in the POD. Therefore, it is more complicated to retrieve the bending angle from the single-frequency data, with little gain in improvement.

Figure 5c,d (page 16, 359): using the vertical/horizontal intervals over which 50 % of the excess occurs seems a reasonable approximate to resolution. However, why is this producing poorer resolution near the surface? Suggestion: why not use 50 % of the ray bending? For example, would Fig 5c,d look the same if you used 50 % of the partial bending angle?

That is an interesting suggestion. We used excess phase rather than bending angle because it is more intuitive in terms of illustrating the distance traveled by the ray as a proxy for horizontal resolution, which is an approximation in any case. It is poorer near the surface because the accumulated excess phase is larger for the longer ray path. The intent is to indicate that if one supposes the part within the central 50 % is representative of the part of the atmosphere that is being measured, the influence of the part outside the central 50 % is being neglected. If there is more accumulated delay on the longer part of the limbs within the troposphere for a tangent point near the surface, then the central 50 % would have to be longer to justify neglecting the influence of the remainder of the raypath.

Here, we use the 50 % excess phase ray path length/thickness as a rough estimate of the scale of the along-path integration. What really matters is the actual "vertical resolution" qualified by the first Fresnel zone with defocusing considered.

Figures 13a and 14 would be easier to interpret if the typical SRO refractivity statistics could be added to the plots, or at least discussed in the text. The ROM SAF monitoring may be useful for this. See https://rom-saf.eumetsat.int/monitoring/index.php

This is a good point. In Figures 13a and 14a, we have added the refractivity statistics between SRO/COSMIC-2 and ERA5 taken from Murphy and Haase 2022, in which only the COSMIC-2 profiles located in the vicinity of the ARs are selected to do the comparison. The website you recommended only has information about bending angles, while this study focuses on refractivity.

Page 32, lines 634-635. "Both of these techniques ...". It reads as if you are saying a local refractivity accounts for horizontal variations along the path. Please revise.

We revised a few sentences here to clarify the local and nonlocal operators. Chen et al. 2018 use a local operator for refractivity and a non-local operator for excess phases. And the non-local and 2-D operators can account for horizontal variations along the path.

"Therefore, the preferred approach is to assimilate refractivity directly, with a local or nonlocal operator (Chen et al., 2018), or bending angle using a 1D or 2D operator (Hordyniec et al., 2024). The non-local and 2D operators take into account variations of atmospheric structure along the long horizontal ray path when assimilating such that the observations can be used in high-resolution models."

**Reviewer #RC2**

The paper is well written describing all the aspects of Airborne Radio Occultation including retrieval. Validation of the Airborne RO data having very good match with dropsonde and comparison with reanalysis demonstrates the high quality observations using ARO. It is worth publishing.

Following are a few specific comments which may be considered and clarified prior to the publication:

Line 107: I agree that dense ARO observations will increase the impact, however it will be good to know the errors of ARO wrt SRO for the lower atmosphere which may increase the overall error in the forecast.

ARO generally has larger errors than SRO. The current ARO system is limited to covering the mid- and upper-troposphere; only about 20-30 % of the profiles penetrate below 2 km, and those profiles generally have downgraded quality. We have dedicated sections to discuss ARO errors and compare them with SRO. In our current data assimilation experiments, we cut off the data at the lowest troposphere to avoid bringing errors to the forecasts.

In Figures 13a and 14a, we have added the refractivity statistics between SRO/COSMIC-2 and ERA5 taken from Murphy and Haase 2022, in which only the COSMIC-2 profiles located in the vicinity of the ARs are selected to do the comparison.

Line 493: Although refractivity anomaly for ARO and ERA analysis looks to be similar in general however there are difference for low and high values.

We do not expect them to match exactly because the ARO profiles are not in the same plane as the transect, and also considering the model resolution (0.25° and 1 hour) and interpolation applied to create the transect. We add this to the text:

"Because the slanted ARO profiles that sample up to 450 km to the side of the flight track are projected onto the plane of the transect, there is expected to be some difference, however the pattern of the ARO observed refractivity closely matches the ERA-5."

Since most of the discussions in this paper are on airborne radio occupation than on atmospheric rivers, move appropriate title can be Airborne radio Occultation System description und its advantages to observe atmospheric river.

This manuscript discusses many ARO features. However, unlike spaceborne RO, which is a global dataset, this ARO dataset is more regionally focused. Many ARO features, such as profile drift, penetrating depth, maximum height, and observation errors, are directly related

to the AR environment and AR Recon flight tracks, which were designed for AR surveys. An important objective of this manuscript is to deliver a comprehensive summary for both modelers and observational scientists interested in AR Recon. Another objective of this manuscript is to provide the background to broaden participation of additional researchers to analyze and assimilate the ARO datasets and contribute to research on atmospheric rivers.

Fig 14 (d): Is there any particular reason for showing mean and SD of Galileo by dashed lines.

This is an oversight; we recreated the figure with the same line style.

Line 609: Statement "highest accuracy between 3 km and 14 km" may need modification as errors till 4 km are 4% and as per Table 4, ARO data is at maximum height of 12.5 km.

This is based on a threshold of 2 % for good quality based on the ARO vs. ERA5 comparisons. We made this explicit in the sentence at line 523:

The current version of the ARO dataset provides the highest accuracy (i.e. better than 2 %) between 3 km and flight level at ~ 14 km

Regarding the 4% and 12.5 km in Table 4, the 4% difference is between ARO and dropsonde, which includes the difference related to spatial separation (up to 500 km) due to the drifting of the ARO profiles, thus is a measure of the variability of the atmosphere rather than the accuracy of the observations. The aircraft's maximum cruise altitude is 14 km, which is also the upper limit for ARO observations. While the dropsondes remove the observations right after they were released from the aircraft for sensors to reach equilibrium, the upper limit is about 12.5 km.

We added the text at line 482:

"The topmost point of the ARO profile is at flight level, whereas usually the topmost dropsonde observations are excluded right after they are released from the aircraft while the sensors reach equilibrium."

In comparison to Satellite RO observations, Airborne is showing higher errors (Table 4). An explanation on the same as well as possible methods to reduce this will be beneficial for the forecasters.

**We added at line 419:**

The overall higher errors of ARO compared to SRO is due to the turbulent motion of the aircraft, such that any error in the velocity estimate of the aircraft introduces a Doppler error in the data before converting to bending angle.

This turbulence-induced noise leads to degraded position accuracy for the aircraft, which subsequently propagates into the final retrieval products. To address this issue, ongoing efforts are focused on improving our processing algorithms to enhance position accuracies. In the interim, we provide the most accurate estimates of the apparent errors associated with ARO products to assist modelers and forecasters in their analyses.

We also added the following at line 563:

"This is expected to provide a dataset with more extensive sampling in the lowest troposphere, as well as a decrease in error with the use of the phase matching bending angle retrieval (Wang et al., 2017)."

Line 250: should be "ionospheric effect" instead of" ionospheric effort"

**Typo corrected.**

Live 261: it will be good to know how much error smoothening introduces?

As stated above, the ARO observation errors/uncertainties are related to the aircraft velocity. The error will propagate into further bending angle and then refractivity retrievals. We apply smoothing to the bending angle based on the theoretical limit of vertical resolution and data sampling rate described in section 2.3. Smoothing reduces the error at the expense of decreasing the vertical resolution. Section 2.3 describes the impact of smoothing on the vertical resolution, and the ability to resolve sharp gradients in bending angle.

**Reviewer #RC3**

First Round of Review of Manuscript AMT-2024-119 Submitted to Atmospheric Measurement Technology

Manuscript Title: Observing Atmospheric Rivers using Multi-GNSS Airborne Radio Occultation: System Description and Data Evaluation

Corresponding Author: Jennifer Haase (jhaase@ucsd.edu)

**Study Summary:**

This manuscript describes the use of airborne GNSS radio occultation for observing atmospheric river (AR) events that impact the west coast of the United States. These airborne radio occultation (ARO) observations are shown to be successful in observing AR events due to the inherent ability of ARO profiles to ignore clouds and precipitation, resulting in data that can observe the thermodynamics of ARs where other remote sensing methods fail due to low vertical resolution or signal attenuation. A ARO full observation and retrieval system is described, and retrieved results were compared to ERA5 model reanalysis output as well as insitu dropsonde observations. Mean refractivity differences between ARO profiles and ERA5/dropsonde profiles was found to be less than 0.5% magnitude above 3 km with varying standard deviation that is higher at lower altitudes, indicating the high quality of the observations and their potential usefulness in numerical weather prediction of AR events.

**General Comments:**

This manuscript is generally well-written and provides very unique data and results. However, my major comment regarding the manuscript is that it is quite long, with more specific provided as part of my comments. Overall, I would recommend publishing this paper after the below comments and suggestions are sufficiently addressed. It is likely that I will not catch all errors, so I would encourage additional read throughs to find any additional issues. My overarching notes for this study are the following:

1. General Readability/Structure and Grammar:

In general, I would avoid the use of phrases like "we implemented" or "our tests" in regard to the experiments that took place. Pronouns are generally not used in technical writing.

The style was adjusted to avoid the active voice and pronouns.

2. Introduction and Motivation:

The introduction is generally well-written with only a few issues to fix detailed in the line-byline comments. Based on a Google Scholar search, the authors do not mention recently published work on engineering of new airborne RO payloads and the use of commercial aircraft for airborne radio occultation (e.g., Chan et al., 2022; Xie et al., 2024). This would be most relevant to the discussion around line 110. Do the authors have a specific reason they chose not to include this potentially relevant information? If not, I would encourage them to work relevant references into their introduction.

We thank the reviewer for pointing out these references, especially the most recent, and we have added them.

**3. Methodology:**

I was not able to find the authors' Equation 1 in (Rüeger, 2002) or (Rüeger, 2002). I am concerned that the authors are not using the correct formulation for refractive index and refractivity. The authors' equation 1 is especially concerning because the second term in Eq. 1 actually removes some of the contribution of water vapor rather than the traditionally-documented additional effects of water vapor on atmospheric refractivity. The authors will need to either fix this error or adequately justify the use of this different equation.

Equation (1) on line 136 had a typo in the first coefficient; we corrected it, and this is the correct version.

$$N = (n-1) \times 10^6 = 77.689 \frac{p}{T} - 6.3938 \frac{p_w}{T} + 3.75463 \times 10^5 \frac{p_w}{T^2}$$

Regarding the difference with Rueger 2002, it was because we combined the  $p_d + p_w = p$ .

$$N = (n-1) \times 10^6 = 77.689 \frac{p_d}{T} + 71.2952 \frac{p_w}{T} + 3.75463 \times 10^5 \frac{p_w}{T^2}$$

Another refractivity equation (Smith and Weintraub 1955) has been extensively used.

$$N = (n-1) \times 10^6 = 77.6 \frac{p}{T} + 3.73 \times 10^5 \frac{p_w}{T^2}$$

We have added the reference for Healy 2011 which shows the difference is ~0.1% in bending angle, and thus even smaller for refractivity. We add one sentence to clarify the potential differences related to different versions of the refractivity formula, at line 138:

"Another refractivity formula frequently used in RO studies is based on (Smith and Weintraub 1953). The two formulas were reported to produce forward modeled bending angle errors of ~0.1% (Healy2011)."

**4. Results and Discussion:**

The first few sections of the results are primarily dedicated to various retrieved profile properties such as resolution, penetration depth, and obliqueness. While I understand the authors' desire to be complete in their analysis, it seems to me that while this information is unique to AR events, this type of information has been described by previous studies that develop the methodology. I would suggest trimming and/or cutting some of the material regarding resolutions, durations, etc. so that the authors can focus more on the observations themselves and their impact on NWP simulations of AR events, as the purpose of the manuscript purports to be focusing on based on the title and abstract.

These features are linked to the AR environment, equipment, and flight tracks. We chose to present them for two main reasons. First, it will provide valuable information for flight planning, to incorporate features such as obliqueness and azimuth dependence into future flight reconnaissance efforts. Second, certain characteristics, including resolution and obliqueness and observation error, are critical in data assimilation and are closely tied to the assimilation operator and its use. While many of these general features are well-known in the context of the specialized field of ARO, our objective is to offer a quantitative assessment. Ultimately, we aim to deliver a comprehensive summary for both scientists who are deploying and designing the reconnaissance flights and the NWP modelers who are using the observations in forecasting models.

The analysis of the effect of the azimuth of the RO relative to the aircraft (e.g, Figure 12) is very interesting. Is the rate of obstruction due to the aircraft something that could be resolved with additional antennae? Or perhaps switching to open-loop tracking? It seems like the highest rates of obstruction occur for the most impactful ARO profiles, so future solutions to this would represent a significant increase in the quality of the ARO data.

The presented results are derived from the closed-loop receiver. While the exact cause is not fully understood, we suspect that signal reception from the front was obstructed by the aircraft fuselage as it flew at an approximate 4° pitch-up angle. Open-loop data recorded from 2021 onward has not yet been processed. We anticipate that penetration depth will improve in all directions; however, this may not fully resolve the asymmetry, with deeper profiles likely remaining concentrated toward the rear.

We agree that investigating alternate locations for the antenna, or adding an additional antenna, for example on the nose or the tail of the aircraft, is worthwhile to maximize the number of profiles. Installing equipment outside the fuselage presents significant challenges, including higher costs and airworthiness considerations, as it requires structural modifications such as cutting holes for the antenna. We are optimistic that such an investigation could be carried out in the future, as the value of the data in NWP forecast improvements is proven.

We have added at line 459:

"Ultimately, a study to optimize the antenna location would be worthwhile to maximize the number of profiles retrieved."

How do the authors correct for errors in bending near the aircraft(receiver)?

The initial bending angle retrieval was smoothed using a variable window width: a wider window (2–5 minutes) was applied for positive elevation bending angles, while a narrower window (31 seconds) was used for negative elevation bending angles below 1 km from the highest impact parameters. Subsequently, the partial bending angle was calculated, followed by the derivation of refractivity. This method has proven effective for 80% of the profiles and is computationally efficient, enabling near real-time processing. To address remaining errors, we have implemented a reprocessing procedure involving human intervention and plan to develop an improved algorithm for automated error correction.

Are the authors down-sampling the ARO profiles to the ERA5 vertical resolution? It is not clear to me, because the authors state they use the pressure-level ERA5 data but then refer to the model grid points in line 593. I am concerned that there would be significant loss of information from the ARO as a result of down-sampling the ARO. Model level data would be more sufficient for this comparison, particularly in the lower troposphere. Additionally, is linear interpolation of the ERA5 temperature and humidity sufficient? Why not use a higher-order interpolation?

The ARO final refractivity profiles are delivered with 100 m sampling following the CDAAC convention. We interpolated the pressure, temperature, and humidity of the ERA5 products into this 100 m grid for comparison. In the earlier years, we used the 37 pressure-level ERA5 provided by NCAR RDA. In a separate study, we analyzed balloon-borne RO data, we noticed evident wave patterns in the differences (ARO-ERA5) above 17 km. We determined these are due to coarse resolution in the pressure-level products not being able to resolve the atmospheric waves. Afterwards, we switched to the 137 native model-level products, effectively reducing the residuals for balloon data.

On the other hand, the difference (model-level vs. pressure-level) for aircraft RO that are mostly below 13 km is very small (same for balloon RO data below 13 km), the mean and STD do not show strong evidence of a systematic problem with the vertical resolution of the pressure level ERA5. There may exist differences in an individual profile, but the statistical mean is very close. Therefore, in this study, what we showed is based on 37 pressure-level ERA5 products and interpolated ERA5 to the ARO data grid.

Regarding the temperature, in the range between 5-14 km, the linear interpolation works fine in the majority of cases. In some instances, the aircraft flew higher (>14 km) and closer to the tropopause. This only happens when the aircraft climbs to a higher cruise altitude after burning most fuel at the end of the flight and is close to the airport, so represents a small number of profiles. However, we appreciate the concern, and in most of our subsequent work, we routinely use the model-level ERA5 to avoid any issues.

I would suggest that the authors do more to put their results in context with other recently published in-atmosphere RO studies, particularly those that occur in high-moisture environments.

Regarding the ARO profiles in high-moisture environments, and in response to the comments below, we offer the following: With the current closed-loop system, we obtain few reliable measurements in the lowest moist troposphere, as most profiles terminate above this moist layer. Only about 20% of the data penetrate below 2 km, and the data quality at these lower levels is generally poor. The current dataset exhibits the highest quality in the mid-to-upper troposphere, where it has been assimilated. We anticipate that the open-loop tracking algorithm will improve the profile penetration depth. Consequently, analyses using a greater number of higher-quality profiles would yield more meaningful insights. However, we added some discussions (attached at the end of this document) on space-borne RO in the moist troposphere and its potential relation to the ARO.

While the ARO data from AR recons undoubtedly provide significant information about the atmospheric state, it is not clear to me how much the ARO profiles, which generally do not penetrate below 4 km due to closed-loop tracking, actually penetrate into the clouds and precipitation resulting from AR. Can the authors comment on what the profiles that penetrate below 4 km show in regard to the regions that are cloudy and/or actively precipitating? I feel like this information would contribute more to the manuscript overall than the long discussion of various resolutions, penetration, etc. of the profiles.

In response to previous comments, we note that with the current closed-loop system, approximately 50% of the profiles penetrate below 4 km, and 20% penetrate below 2 km. The NOAA G-IV primarily operated in the upstream region of an AR, which, while moist, experienced minimal active precipitation. Our analysis of the correlation between penetration depth and the moist environment revealed no significant differences between colder, higher-latitude locations and warmer, lower-latitude regions, or between the warm and cold sectors of an AR. Instead, engineering factors, such as the aircraft heading, related to the fuselage blocking, as illustrated in Fig. 12, appear to have a stronger influence, overshadowing any potential relationship with the moist environment. Over the four years of experimental deployment presented in this study, the GNSS receiver and antenna underwent several upgrades, introducing various engineering factors that affect the performance of the current closed-loop system. We aim to utilize open-loop data to investigate these issues more thoroughly.

**5. Conclusions:**

Does the AR dataset have a formal citation? What about a DOI? Archiving the data with NOAA would allow for the creation of both of these things fairly easily. The data should have a proper citations regardless, but archiving with NOAA is merely a suggestion based on reviewer experience.

**A DOI has been created with the UCSD library research data curation service to deposit the final ARO data for the archive. The DOI will be included in the manuscript before final publication.**

Again, the comparisons of the ARO profiles with dropsonde sand ERA5 are really not properly put into context with more recent studies. I strongly suggest that the authors do a more thorough review of recently published work to compare their statistics with other observations from SRO and ARO, particularly if high-moisture observations are available.

The scope of this study is to present a dataset and provide helpful information for modelers and aircraft operators, including insights into observation errors relevant to data assimilation and modeling efforts. The comparison between dropsonde and ARO data highlights the complementary nature of ARO, which samples a broader and different environment than dropsondes. We also plan to conduct an ARO vs. SRO comparison in a separate case study, as the spatiotemporal colocation of these observations significantly limits the sample size. It is important to note that high-moisture observations are not a primary strength of the current dataset; however, we anticipate that future open-loop datasets will help address these limitations. We added some discussions (attached at the end of this document) on spaceborne RO in the moist troposphere and its potential relation to the ARO.

Please see my line-by-line comments for more specific details.

**Line-by-Line Comments:**

1. L013: "Lock" is a little bit general. I assume the authors mean signal phase lock. Please consider referring specifically to signal phase lock.

We intend to keep it general about the GNSS signal being tracked, so we changed it to "signal tracking."

".... below which the receiver loses or cannot initiate signal tracking."

2. L026-027: Please reword this sentence to remove repeated occurrences of "on the other hand" within the same sentence. I would suggest at least two sentences from this one.

These two sentences are reworded for clarification.

"They can provide much-needed precipitation to support water supply and alleviate droughts, but prolonged heavy rainfall from ARs can also lead to severe flooding, causing fatalities and significant economic losses."

3. L144: I would advocate the use of an Oxford comma here. Specifically, after "transmitter (satellite)..."

**Added**

4. L241: There should be an Oxford comma after "and" here for lists.

**Added**

5. L249: "...relativity effect" should be "relativity effects" here.

**Corrected.**

6. L250: Should "effort" be "effects" here? Also, is this simply the first-order ionospheric correction? Was any testing done using higher-order corrections? If not, what would be the expected effects, if any?

**The typo is corrected.**

Only the first-order ionospheric correction was applied, which was limited by the precise point positioning (PPP) software and our time domain processing approach. Making the first order ionosphere correction is tricky at the zero elevation angle point, and is an area of active research because it could improve the retrievals near the aircraft height. That would be expected to provide a more significant improvement than the higher order correction. We recorded full GNSS frequencies, and many GPS satellites broadcast L5 signals, which provide extra information for ionospheric correction.

7. L262-263: I would specify that the "positive bending angle" and "negative bending angle" are really "positive elevation angle bending" and "negative elevation angle bending" (or something similar) throughout the paper. Otherwise, it implies that the bending angle itself is negative, which is not physically consistent. This should be true throughout the manuscript.

Thanks for pointing this out. We have corrected the "positive/negative bending" to "positive/negative elevation bending."

8. Figure 3: What is the reason for the color shading? The link to the AR Recon data in the caption of Figure 3 should be formatted into a proper citation as a dataset to be consistent with AMT regulations.

The color shadowing only separates two adjacent years on the figure. We added the citation for the link; it was also included in the "data availability" section.

9. L332-333: The description of the colored lines in Figure 4 should probably be limited to the figure caption.

The sentences about line colors are removed from the text and added into the figure caption.

Black lines indicate the flight tracks and red and blue lines denote setting and rising occultations, respectively.

10. L350: I don't believe that "decimated" is the correct word here. Perhaps the authors meant "delineated"?

"Decimated" here means down-sampled. We follow the terminology used by GPS utility software teqc, which has an option of decimating (-O.dec) the high-rate data to lower the sampling rate.

11. L404: I would be careful describing this as "tomographic-style". I'm not entirely convinced that this would fit the traditional tomography definition.

We removed "tomographic-style".

12. L527: I would change the end of this sentence to be " ...starboard and port directions, respectively."

**Added.**

13. L545: "co-located" should be "colocated" to be consistent with journal hyphenation rules and consistency. Typically, "collocated" or "colocated" are the journal-recognized spellings. This should be changed throughout the manuscript.

The instances of "co-located" used in section 4 are corrected.

14. L550-552: Please re-word these sentences to remove the second sentence. Perhaps something like "... solely from the G-IV, resulting in a much more extensive dataset..."

These sentences are reworded into one for simplicity.

"Since 2020, more than 600 ARO profiles have been retrieved annually, alongside 400 to 500 dropsondes deployed from the G-IV, forming an extensive dataset that supports comprehensive and robust statistical analysis to evaluate data quality."

15. L554: "dropsonde" should be plural here

Corrected.

16. L556: "ARO profiles ... from the track" is not needed as it was heavily discussed in previous sections.

The whole sentence was removed.

ARO profiles were retrieved along the entire flight track, however they are irregularly distributed and slant away from the track.

17. L560: I think there is an indent here where there shouldn't be one.

We did not locate any anomaly in the raw latex files to create such an indent at the beginning. It is likely due to the auto line spacing adaptation. We will defer to type editors to fix this.

18. L596: Please see general comment #2a regarding the calculation of atmospheric refractivity. This is likely creating errors due to the incorrect refractivity formulation.

We corrected the error in the formula to be the correct value that was used throughout our study; extra clarifications are provided in comment #2a.

19. Table 4: I would suggest replacing the "nan" values with something to indicate that there is simply no data there. Maybe a dash?

The "nan" in the table is replaced by a long dash. Extra notes are added in the caption.

**Reviewer References:**

Chan, B. C., Goel, A., Kosh, J., Reid, T. G. R., Snyder, C. R., Tarantino, P. M., Soedarmadji, S., Soedarmadji, W., Nelson, K., Xie, F., and Vergalla, M.: Commercial GNSS Radio Occultation on Aerial Platforms With Off-The-Shelf Receivers, navi, 69, navi.544, https://doi.org/10.33012/navi.544, 2022.

Rüeger, J. M.: Refractive Index Formulae for Radio Waves, XXII International Federation of Surveyors, Washington, DC, USA, Integration of Techniques and Corrections to Achieve Accurate Engineering Survey, 13, 2002.

Rüeger, Jean M.: Refractive Indices of Light, Infrared, and Radio Waves in the Atmosphere, University of New South Wales, 2002.

Xie, F., Nelson, K. J., Chan, B. C., Goel, A., Kosh, J., and Vergalla, M.: First Results of Airborne GNSS Radio Occultation Sounding From Airbus Commercial Aircraft, Geophysical Research Letters, 51, https://doi.org/10.1029/2024GL110194, 2024.

The following is added discussions (in section 4.3) about SRO in the moist troposphere and relation with ARO.

[revised manuscript text omitted]

---

## Referee Report (RR1)

**Second Round of Review of Manuscript AMT-2024-119 Submitted to Atmospheric Measurement Technology**

Manuscript Title: Observing Atmospheric Rivers using Multi-GNSS Airborne Radio Occultation: System Description and Data Evaluation

Corresponding Author: Jennifer Haase (jhaase@ucsd.edu)

**Study Summary:**

This manuscript describes the use of airborne GNSS radio occultation for observing atmospheric river (AR) events that impact the west coast of the United States. These airborne radio occultation (ARO) observations are shown to be successful in observing AR events due to the inherent ability of ARO profiles to ignore clouds and precipitation, resulting in data that can observe the thermodynamics of ARs where other remote sensing methods fail due to low vertical resolution or signal attenuation. A ARO full observation and retrieval system is described, and retrieved results were compared to ERA5 model reanalysis output as well as in-situ dropsonde observations. Mean refractivity differences between ARO profiles and ERA5/dropsonde profiles was found to be less than 0.5% magnitude above 3 km with varying standard deviation that is higher at lower altitudes, indicating the high quality of the observations and their potential usefulness in numerical weather prediction of AR events.

**General Comments:**

This manuscript has improved after all reviewers' comments and the authors' modifications, and I appreciate the authors' thorough responses to my comments. I would encourage the authors to add some of the information in their responses as text to the manuscript where relevant. The manuscript is still extremely valuable for its unique dataset and the information obtained from it. I think the decision to move the ARO processing description to a set of appendices was a good choice. I would also encourage additional read throughs to find any additional grammatical issues prior to publication.

**Given the state of the manuscript, I have only minor comments. I recommend publishing this paper after the below comments are sufficiently addressed.**

Please see my line-by-line comments for more specific details.

**Line-by-Line Comments:**

1. L011: I suggest removing the parentheticals surrounding the flight hours and using something like: "... obtained from 39 flights over approximately 260 flight hours by tracking multiple GNSS constellations." Not required, but I think it might help the sentence read better.

- 2. L027: Remove "but" from the "but prolonged heavy rainfall..." sentence, it doesn't make sense there.
- 3. L035: I recommend ending this sentence with "... dense horizontal sampling." and creating a new sentence with "However, ..." here.
- 4. L086: I am not sure if you want to specify COSMIC as COSMIC-1 or not in the manuscript. You may also provide the full name for the mission and a citation of Anthes et al., (2008) here since it is the first mention of the mission in the manuscript.
- 5. L105-106: I would encourage the authors to at least provide a citation for COSMIC-2 here, such as Schreiner et al., (2020).
- 6. Figure 3 Caption: "... and Galileo constellations, respectively."
- 7. L395: Here the authors use "COSMIC-1" but in other places the authors use "COSMIC". Please make sure the references to COSMIC-1 are consistent in the manuscript. Personally, because COSMIC-2 has launched, I would advocate for "COSMIC-1".

**Reviewer References:**

Anthes, R. A., Bernhardt, P. A., Chen, Y., Cucurull, L., Dymond, K. F., Ector, D., Healy, S. B., Ho, S. P., Hunt, D. C., Kuo, Y. H., Liu, H., Manning, K., McCormick, C., Meehan, T. K., Randel, W. J., Rocken, C., Schreiner, W. S., Sokolovskiy, S. V., Syndergaard, S., Thompson, D. C., Trenberth, K. E., Wee, T. K., Yen, N. L., and Zeng, Z.: The COSMIC/FORMOSAT-3 Mission: Early Results, Bulletin of the American Meteorological Society, 89, 313–334, https://doi.org/10.1175/bams-89-3-313, 2008.

Schreiner, W. S., Weiss, J. P., Anthes, R. A., Braun, J., Chu, V., Fong, J., Hunt, D., Kuo, Y.-H., Meehan, T., Serafino, W., Sjoberg, J., Sokolovskiy, S., Talaat, E., Wee, T. K., and Zeng, Z.: COSMIC-2 Radio Occultation Constellation: First Results, Geophysical Research Letters, 47, https://doi.org/10.1029/2019gl086841, 2020.

---

## Author Response (AR2)

We appreciate the additional comments from both reviewers. We carefully addressed them, with comments provided here and modifications made in the manuscript.

In the following sections, the reviewer's original comments are in black, our comments are in blue, and the corresponding revisions are in red.

**Reviewer #2 mentioned the issues in line 288.**

We adjust the sentence to emphasize how one profile is formed. The "flight level" and "surface" are not representative; they are replaced with "upward" and "downward" to describe the motion direction of the ray path.

One complete profile is formed when the signal ray paths connecting the satellite transmitter and the aircraft receiver traverse the atmosphere downward (upward) during a setting (rising) occultation.

**Reviewer #1:**

Second Round of Review of Manuscript AMT-2024-119 Submitted to Atmospheric Measurement Technology Manuscript Title: Observing Atmospheric Rivers using Multi-GNSS Airborne Radio Occultation: System Description and Data Evaluation

Corresponding Author: Jennifer Haase (jhaase@ucsd.edu)

**Study Summary:**

This manuscript describes the use of airborne GNSS radio occultation for observing atmospheric river (AR) events that impact the west coast of the United States. These airborne radio occultation (ARO) observations are shown to be successful in observing AR events due to the inherent ability of ARO profiles to ignore clouds and precipitation, resulting in data that can observe the thermodynamics of ARs where other remote sensing methods fail due to low vertical resolution or signal attenuation. A ARO full observation and retrieval system is described, and retrieved results were compared to ERA5 model reanalysis output as well as insitu dropsonde observations. Mean refractivity differences between ARO profiles and ERA5/dropsonde profiles was found to be less than 0.5% magnitude above 3 km with varying standard deviation that is higher at lower altitudes, indicating the high quality of the observations and their potential usefulness in numerical weather prediction of AR events.

General Comments:

This manuscript has improved after all reviewers' comments and the authors' modifications, and I appreciate the authors' thorough responses to my comments. I would encourage the authors to add some of the information in their responses as text to the manuscript where relevant. The manuscript is still extremely valuable for its unique dataset and the information obtained from it. I think the decision to move the ARO processing description to a set of appendices was a good choice. I would also encourage additional read throughs to find any additional grammatical issues prior to publication. Given the state of the manuscript, I have only minor comments. I recommend publishing this paper after the below comments are sufficiently addressed. Please see my line-by-line comments for more specific details.

**Line-by-Line Comments:**

L011: I suggest removing the parentheticals surrounding the flight hours and using something like: "... obtained from 39 flights over approximately 260 flight hours by tracking multiple GNSS constellations." Not required, but I think it might help the sentence read better.

**Corrected.**

L027: Remove "but" from the "but prolonged heavy rainfall..." sentence, it doesn't make sense there.

**'but' removed.**

L035: I recommend ending this sentence with "... dense horizontal sampling." And creating a new sentence with "However, ..." here.

The long sentence is broken into two sentences. The added new sentence is below:

However, these observations often have poor vertical resolution.

L086: I am not sure if you want to specify COSMIC as COSMIC-1 or not in the manuscript. You may also provide the full name for the mission and a citation of Anthes et al., (2008) here since it is the first mention of the mission in the manuscript.

In the manuscript, we use COSMIC, rather than COSMIC-1, to match how it is called in the cited references, where detailed descriptions can be found. We added some clarifications to avoid the possible confusion about COSMIC-1 and COSMIC.

The most notable SRO mission, COSMIC, also known as COSMIC-1, was launched in 2006 and provided many RO observations \citep{Anthes2008}.

L105-106: I would encourage the authors to at least provide a citation for COSMIC-2 here, such as Schreiner et al., (2020).

**Recommended reference is added.**

Figure 3 Caption: "... and Galileo constellations, respectively."

**Added.**

L395: Here the authors use "COSMIC-1" but in other places the authors use "COSMIC". Please make sure the references to COSMIC-1 are consistent in the manuscript. Personally, because COSMIC-2 has launched, I would advocate for "COSMIC-1"

We choose to keep COSMIC, as it is named in the cited references. To avoid confusion, we added a one-sentence description at the beginning:

The most notable SRO mission, COSMIC, also known as COSMIC-1, was launched in 2006 and provided many RO observations \citep{Anthes2008}.

**Reviewer References:**

Anthes, R. A., Bernhardt, P. A., Chen, Y., Cucurull, L., Dymond, K. F., Ector, D., Healy, S. B., Ho, S. P., Hunt, D. C., Kuo, Y. H., Liu, H., Manning, K., McCormick, C., Meehan, T. K., Randel, W. J., Rocken, C., Schreiner, W. S., Sokolovskiy, S. V., Syndergaard, S., Thompson, D. C., Trenberth, K. E., Wee, T. K., Yen, N. L., and Zeng, Z.: The COSMIC/FORMOSAT-3 Mission: Early Results, Bulletin of the American Meteorological Society, 89, 313–334, https://doi.org/10.1175/bams-89-3-313, 2008.

Schreiner, W. S., Weiss, J. P., Anthes, R. A., Braun, J., Chu, V., Fong, J., Hunt, D., Kuo, Y.-H., Meehan, T., Serafino, W., Sjoberg, J., Sokolovskiy, S., Talaat, E., Wee, T. K., and Zeng, Z.: COSMIC-2 Radio Occultation Constellation: First Results, Geophysical Research Letters, 47, https://doi.org/10.1029/2019gl086841, 2020.